# The neglected role of abandoned cropland in supporting both food security and climate change mitigation

Qiming Zheng [1,2] ✉, Tim Ha[2], Alexander V. Prishchepov [3,4], Yiwen Zeng [2,5], He Yin [6] & Lian Pin Koh [2] ✉

Despite the looming land scarcity for agriculture, cropland abandonment is widespread globally. Abandoned cropland can be reused to support food security and climate change mitigation. Here, we investigate the potentials and trade-offs of using global abandoned cropland for recultivation and restoring forests by natural regrowth, with spatially-explicit modelling and scenario analysis. We identify 101 Mha of abandoned cropland between 1992 and 2020, with a capability of concurrently delivering 29 to 363 Peta-calories yr$^{-1}$ of food production potential and 290 to 1,066 MtCO$_2$ yr$^{-1}$ of net climate change mitigation potential, depending on land-use suitability and land allocation strategies. We also show that applying spatial prioritization is key to maximizing the achievable potentials of abandoned cropland and demonstrate other possible approaches to further increase these potentials. Our findings offer timely insights into the potentials of abandoned cropland and can inform sustainable land management to buttress food security and climate goals.

Global cropland expansion has been going hand-in-hand with rapid population growth and food demand increase during the last 300 years[1,2]. Despite overall progress on yield improvements and functioning of agriculture value chains for the last decades[3], by 2021, around 702 to 828 million people, or 9.8% of the world population, were still undernourished[4]. Although agricultural intensification has the potential to reduce the pressure of expanding cropland, the ongoing population growth and increasing per capita food consumption suggest future cropland expansion is inevitable. Recent projections with the Shared Socioeconomic Pathways (SSPs) scenario framework estimated a total cropland expansion of 25 to 226 Mha over the next three decades, from the most to the least sustainable scenarios[5,6]. Future expansion of cropland would put areas with high environmental values and biodiversity-rich frontiers at risk[7–10].

At the same time, the food sector has been contributing significantly to climate change, accounting for about one-third of total anthropogenic GHG emissions[11]. Land management measures that reduce emissions from the food sector are a vital part of international efforts to mitigate climate change. Along with other nature-based climate solutions (e.g., reforestation), these measures represent a large proportion of mitigation actions to keep the global average temperature increase below 2 °C and have been integrated into many countries' National Determined Contributions (NDCs) under the Paris Agreement[12,13]. However, deploying these climate change mitigation measures at scale to limit global warming below 2°C would require large swaths of land (e.g., 145-250 Mha net forest growth by 2050 projected by SSPs-26 scenarios), thereby putting these measures into conflict with future cropland expansion[14].

While the looming land scarcity for agriculture and the intensifying conflicts in land-use demand, cropland abandonment is widespread globally as a result of land degradation, institutional and socioeconomic changes, disasters, armed conflict, and

[1]Department of Land Surveying and Geo-Informatics, Hong Kong Polytechnic University, Hung Hom, Kowloon, Hong Kong. [2]Centre for Nature-based Climate Solutions, National University of Singapore, Singapore 117546, Singapore. [3]Department of Geosciences and Natural Resource Management (IGN), University of Copenhagen, Øster Voldgade 10, DK-1350 København K, Denmark. [4]Center for International Development and Environmental Research (ZEU), Justus Liebig University, Senckenbergstraße 3, 35390 Giessen, Germany. [5]School of Public and International Affairs, Princeton University, Princeton, NJ 08544, USA. [6]Department of Geography, Kent State University, Kent, OH 44242, USA. ✉e-mail: qiming.zheng@polyu.edu.hk; lianpinkoh@nus.edu.sg

urbanization[15–17]. Estimates from satellite imagery show that along with 217 Mha of gross cropland expansion, extensive gross cropland abandonment (79 Mha) took place globally from 2003 to 2019[1]. Cropland abandonment is predicted to continue in abandonment hotspots, such as Europe, Russia, Central and East Asia, and the Americas[5,18]. While cropland abandonment gives rise to threats to food security, recent studies and land management projects (e.g., European Union's Rural Development Programme) highlighted the feasibility of abandoned cropland to be cultivated again for supplying food production (recultivation) or to be restored to natural habitats via natural growth (reforestation) for enhancing carbon sequestration[19–24].

It remains unclear regarding how recultivating and reforesting global abandoned cropland can support food production and climate change mitigation, respectively, and how they are pertinent to achieve food security and climate goals[25]. Assessments of trade-offs and synergies across different purposes of using abandoned cropland are also generally lacking. In most cases, assessment is based on a sole purpose in mind (e.g., recultivation only)[26,27]. Alternative purposes are insufficiently considered, even though the area of interest is more suitable to serve other purposes[28]. At the same time, it has been recently reported that abandonment can be ephemeral, swinging between different land-use purposes, e.g., between producing food and sequestering carbon[18,29]. Thus, this underscores the importance of understanding the potential and trade-offs between different purposes of using abandoned cropland and calls for a better land-use management strategy for abandoned cropland. In addition, given the spatial variability in crop yield and carbon sequestration rates across the globe, spatially prioritized land management strategies are essential in helping various stakeholders seek the most beneficial and optimum solutions[18,26]. Understanding the benefit of applying spatially prioritized strategies is essential to best achieve the potential of abandoned cropland and craft win-win synergies.

Here, we assess the potential and trade-offs of using abandoned cropland for food production and climate change mitigation based on geospatial modelling, machine learning, and scenario simulation approaches. We aim to address three research questions: (1) how much global abandoned cropland is suitable for recultivation and reforestation via natural regrowth, respectively, and what is the resulting potential for food production and climate change mitigation? (2) how to balance the trade-offs between these two purposes? (3) how to maximize their synergies, as well as the achievable potentials? Collectively, our analysis offers timely insights into the untapped potential of abandoned cropland that can inform land-use decision-making processes and support policymakers in meeting ambitious food security and climate goals from land-use sectors.

## Results

### Global abandoned cropland

Based on ESA-CCI land-cover time series data and FAO's definition of cropland abandonment, we identify that 101 Mha (66-136 Mha, 95% confidence intervals) of cropland was abandoned between 1992 and 2020−equivalent to an average cropland abandonment rate of 3.6 Mha yr[-1] (see Methods). The extent of abandoned cropland corresponds to 7.0% and 6.4% of active cropland areas in 1992 and 2020, respectively. This finding confirms cropland abandonment is prevailing in Asia (33 Mha), Europe (22 Mha), and Africa (19 Mha), particularly in Russia (12.4 Mha), China (8.7 Mha) and Brazil (8.4 Ma) (Fig. 1a).

Following the well-established methodology of assessing land cover change mapping accuracy[30], we report the overall accuracy (83%), user's accuracy (95%), producer's accuracy (77%), and F1 score (0.85) of our abandoned cropland map from 1992 to 2020, which are within the range of mapping accuracies of previous studies[31,32] (see Methods). Besides, a qualitative comparison against the other 42 studies with different abandonment identification approaches, input datasets, and study periods shows that the spatial pattern of our

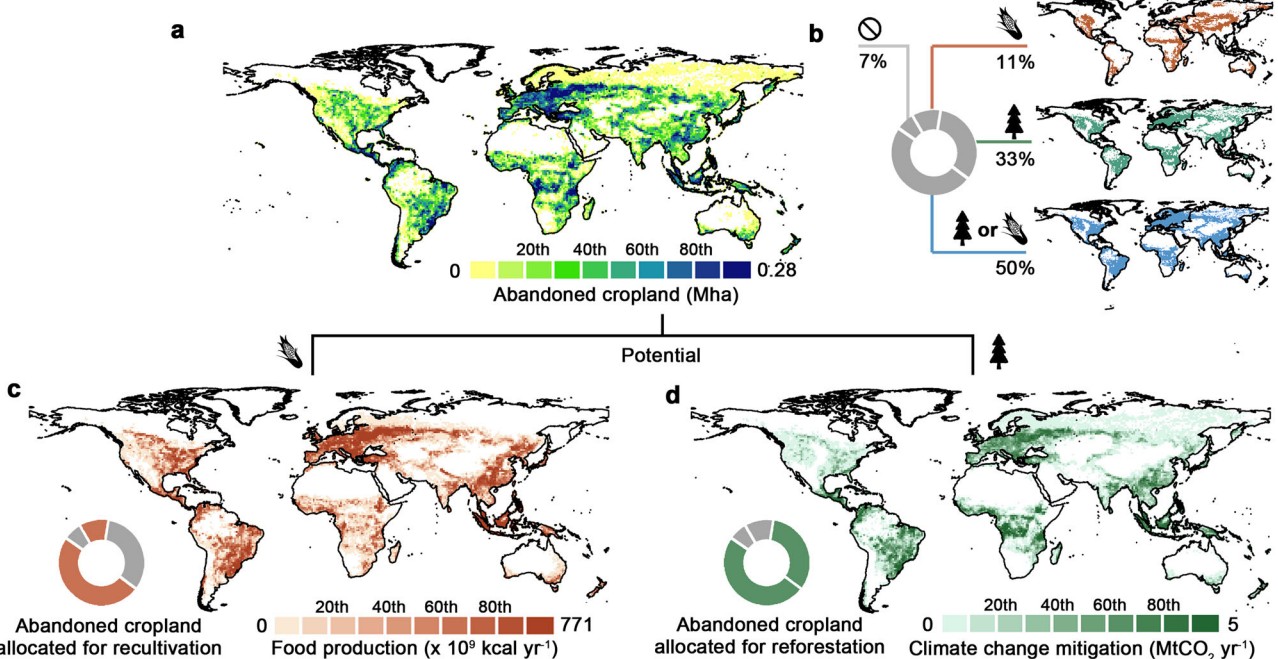

**Fig. 1 | Global abandoned cropland: extent, suitability, and potential.** Abandoned cropland extent (**a**) and binary maps of suitable area for recultivation and reforestation (**b**). Proportions (donut chart) and spatial distribution (maps) of four suitability types of abandoned cropland: only suitable for either recultivation (11%) or reforestation (33%), suitable for both purposes (50%), and not suitable for both purposes (7%). Food production potential (**c**) and climate change mitigation

potential (**d**) when either recultivation or reforestation is solely considered. The colored areas in the donut charts illustrate which suitability types and proportions of abandoned cropland are allocated for either recultivation (brown) or reforestation (green). The color scales are categorized by every 20th quantile, and the maps are aggregated from 10 arc-second to 1 arc-degree resolution for better visualization (similarly hereinafter).

identified cropland abandonment is well consistent with previous studies, such as the abandonment due to socioeconomic and institutional changes (collapse of former Soviet Union)[33], regional conflicts (Central Africa)[34], biophysical constraints (poor soils Switzerland, rugged terrain in Nepal, and limited water supply in Southern Europe)[35,36], and shifts in land-use policy (Thailand)[37] (Supplementary Data 1). Our identified abandonment pattern is particularly well consistent with studies relying on the same input dataset[38,39], where the differences are mainly caused by the study period.

### Food production and climate change mitigation potential

Accounting for biophysical, agro-climatic, and socioeconomic factors, we trained a machine learning model with abandoned cropland that had been recultivated and mapped abandoned cropland suitable for recultivation. We find 61 Mha of abandoned cropland is suitable for recultivation but has not been recultivated between 1992 and 2020 (Fig. 1b; Supplementary Fig. 1). We estimated the calorie food production potential by recultivating suitable abandoned cropland with 15 major food crops based on their present-day productivities, and with food waste and loss deducted and protected areas excluded (see Methods). If all suitable abandoned cropland is recultivated, it would yield an additional food supply of 363 Pcal yr$^{-1}$ (Peta-calories=$10^{12}$ kcal), with a potential to feed 292 to 476 million people per year based on different diets and calorie intake conditions (Fig. 1c; Supplementary Text 1). We also accounted for emissions from clearing the historically accumulated aboveground biomass carbon for recultivation by assuming all biomass is immediately oxidized (see Methods). For example, clearing all the suitable abandoned cropland land for recultivation will lead to an instant emission of 4.7 GtCO$_2$ or a 30-year average emission of 156 MtCO$_2$ yr$^{-1}$ (Supplementary Text 2).

We identified 83 Mha of abandoned cropland suitable for natural forest regrowth with a recently published map presenting areas suitable to return to native forest cover preceding human disturbance based on biophysical, climatic, and lithological conditions[40]. We then estimated the net climate change mitigation potential by considering the achievable amount of carbon sequestration from natural reforestation and the emissions from land clearing for recultivation (see Methods). If all suitable abandoned cropland is naturally reforested, it could sequester an amount of carbon by 1,080 MtCO$_2$ yr$^{-1}$, thus making it an important but missed opportunity to accelerate efforts to mitigate climate change (Fig. 1b & 1d; Supplementary Fig. 1). By comparison, this climate change mitigation potential can help to meet, on average, 17% of the emission reduction targets that 120 countries have committed to their unconditional NDCs, while a great variation is also observed among countries, e.g., USA (0.4%), Indonesia (19%) and Ethiopia (49%) (Supplementary Data 2). The potential also equals 3-7% of the global emission reduction needed for achieving the 2°C climate goal (SSPs-26, 2020-2050)[6]. This does not undermine the contribution

of reforesting abandoned cropland, but it reflects the fact that like other natural climate solutions, only serves as an essential complement but cannot override the imperative for mitigation from the energy and industry sector[41].

### Trade-offs and synergies

Of the 61 Mha and 83 Mha of abandoned cropland suitable for recultivation or reforestation, we find 50 Mha is suitable for both purposes (Fig. 1b). To investigate the trade-offs between these two purposes and how to maximize synergies, we simulated a series of scenarios, where each pixel of abandoned cropland was spatially allocated to either recultivation or reforestation based on its suitability (see Methods).

When targeting using all suitable abandoned cropland, we find the food production potential is maximized in a scenario termed "maximizing food production", where 61 Mha of abandoned cropland would be recultivated while 33 Mha would be reforested (Table 1). On the other hand, in the scenario where climate change mitigation is maximized (herein, we use "maximizing climate change mitigation"), 83 Mha of abandoned cropland would be allocated to reforestation and 11 Mha to recultivation. These two scenarios represent two extremes of achievable food production and climate change mitigation potential. All the other intermediate scenarios are situated between these two scenarios, with varying amounts of abandoned cropland allocated for each purpose and, thus, varying degrees of achievable food production and climate change mitigation potential. As a result, we find synergies between both purposes vary greatly over space and the achievable potential (Fig. 2 and Table 1). The estimated food production potential on abandoned cropland varies eightfold across these scenarios, from 29 Pcal yr$^{-1}$ to 363 Pcal yr$^{-1}$, whilst the climate change mitigation potential ranges from 290 MtCO$_2$ yr$^{-1}$ to 1,066 MtCO$_2$ yr$^{-1}$ (see a visualization of all simulated outcomes in Supplementary Fig. 2).

To compare these simulated outcomes, we established a hypothetical indicator – combined potential (sum of achieved food production and climate change mitigation potential in percentage) – to assess the integrated benefit under each scenario. Among all scenarios, the "maximizing combined potential" scenario allocates abandoned cropland pixels to the optimal purpose by considering the comparative advantage of each pixel based on its cropland productivity and carbon sequestration rates. This scenario yields the highest combined potential (143%), concurrently achieving 79% of maximum food production potential and 72% of maximum climate change mitigation potential (Fig. 2d and Table 1). Notably, its combined potential is up to 36% higher than other scenarios, such as the "maximizing climate change mitigation" scenario (107%) and "equal allocation" scenario (121%). The prominent difference is due to high spatial heterogeneity in cropland productivity and carbon sequestration rates across the globe (Supplementary Fig. 3). Taking the

**Table 1 | The amount of abandoned cropland allocated for recultivation and reforestation and the corresponding food production potential and climate change mitigation potential of four representative scenarios (see Methods for detailed scenario narratives)**

| Representative scenarios | Abandoned cropland for recultivation Mha (% of max.)* | Abandoned cropland for reforestation Mha (% of max.) | Food production potential Pcal yr$^{-1}$ (% of max.; A) | Net climate change mitigation potential MtCO$_2$ yr$^{-1}$ (% of max.; B) | Combined potential (A + B)** |
|---|---|---|---|---|---|
| Maximizing food production | 61 (100%) | 33 (39%) | 363 (100%) | 290 (27%) | 127% |
| Maximizing climate change mitigation | 11 (18%) | 83 (100%) | 29 (8%) | 1066 (99%) | 107% |
| Equal Allocation | 47 (77%) | 47 (56%) | 269 (74%) | 508 (47%) | 121% |
| Maximizing combined potential | 43 (70%) | 51 (61%) | 295 (81%) | 667 (62%) | 143% |

*: Values in parentheses specify the percentage of abandoned cropland allocated, for either recultivation or reforestation, out of the total suitable extent for recultivation (61 Mha, 100%) and reforestation (83 Mha, 100%) or the percentage of the achieved potential out of the maximum achievable food production potential (363Pcal yr$^{-1}$, 100%) and climate change mitigation potential (1,080 MtCO$_2$ yr$^{-1}$, 100%). **: The combined potential (A + B; %) is calculated by the sum of achieved food production potential (A; %) and climate change mitigation potential (B; %), which serves as a hypothetical indicator to evaluate the integrated outcome of each scenario.

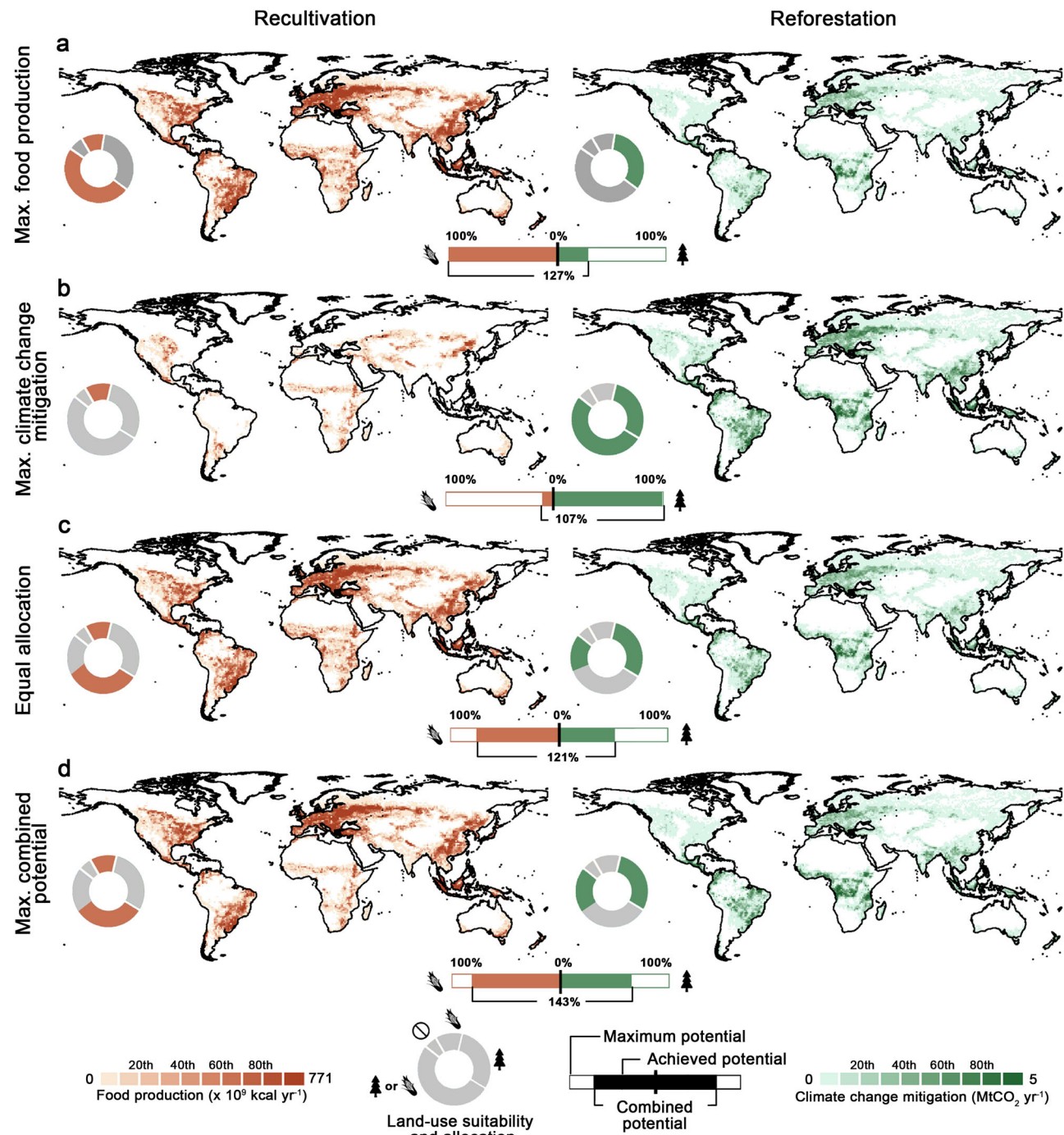

**Fig. 2 | Trade-offs between recultivating and reforesting abandoned cropland.** Global food production and climate change mitigation potential of abandoned cropland under four representative scenarios: "maximizing food production" scenario (**a**), "maximizing climate change mitigation" scenario (**b**), "equal allocation" scenario (**c**), and "maximizing combined potential" scenario (**d**). The donut charts illustrate four suitability types of abandoned cropland and their proportions among the total amount of abandoned cropland, while the colored areas indicate how abandoned cropland is allocated for either purpose. The colored horizontal bars show the percentage of the achieved potential out of the maximum achievable food production potential (363 Pcal yr$^{-1}$, 100%) and climate change mitigation potential (1,080 MtCO$_2$ yr$^{-1}$, 100%), respectively.

"maximizing food production" scenario as an example, despite achieving the maximum food production potential, a considerable proportion of low-productivity areas are allocated for recultivation, e.g., in Central Africa, while these areas have a comparatively higher potential for sequestering carbon[42]. By comparison, the "maximizing combined potential" scenario best leverages the comparative advantage of abandoned cropland pixel, thus maximizing the synergies between recultivation and reforestation.

## Benefits of spatial prioritization

The remarkable difference in the simulated outcomes across space and scenarios underscores the necessity for applying spatially prioritized land management strategies to make full use of the potential in abandoned cropland. It is also very pertinent to land management because, due to limited resources, most real-world efforts (e.g., area-based initiatives[43]) only target recultivating or reforesting a certain amount of suitable areas rather than all of them,

thus making knowing where to prioritize and the benefits of prioritization imperative.

We assessed the impact of applying a spatially prioritized allocation strategy on the resulting potential of abandoned cropland by comparing outcomes simulated with and without spatial prioritization (i.e., randomized allocation) across different percentages of abandoned cropland that are used. We find spatial prioritization can increase food production and climate change mitigation potential by up to 59% and 43% (Fig. 3). Such benefit is most pronounced when only a portion of global abandoned cropland is used. For example, when 30% (30.3 Mha) of abandoned cropland is used, applying spatially-prioritized allocation produces an additional 29% ("maximizing food production" scenario) and 19% ("maximizing combined potential" scenario) of food over scenarios without prioritization, while increasing the total carbon sequestration by 31% ("maximizing climate change mitigation" scenario) and 27% ("maximizing combined potential" scenario). Applying spatial prioritization improves land-use efficiency. To achieve the same amount of food production and climate change mitigation potential, scenarios with spatial prioritization consume, on average, 14% (recultivation) and 10% (reforestation) lesser abandoned cropland than those without. Under "maximizing food production" scenario, to achieve 200 Pcal yr$^{-1}$ food production potential, applying randomized allocation would require 33 Mha of abandoned cropland, but only require 19 Mha of abandoned cropland (74% lesser than randomized allocation).

The frequency distributions of cropland productivity and carbon sequestration rates in used abandoned cropland areas explain how spatial prioritization achieves these enhanced outcomes (Fig. 3). Applying spatially prioritized allocation ensures recultivation and reforestation efforts focus on areas with high cropland productivity or carbon sequestration rate. When using 30% of global abandoned cropland under the "maximizing combined potential" scenario, for instance, spatially prioritized abandoned cropland exhibits average cropland productivity of $7.7 \times 10^6$ kcal yr$^{-1}$ ha$^{-1}$ and carbon sequestration rate of 11.8 MgCO$_2$ yr$^{-1}$ ha$^{-1}$, approximately three times higher than the corresponding scenarios without spatial prioritization ($1.9 \times 10^7$ kcal yr$^{-1}$ ha$^{-1}$ and 3.6 MgCO$_2$ yr$^{-1}$ ha$^{-1}$, respectively). Our further analysis indicates that recultivation or reforestation of abandoned cropland spatially prioritized in the following regions could yield the best food production and climate change mitigation outcomes: Central and

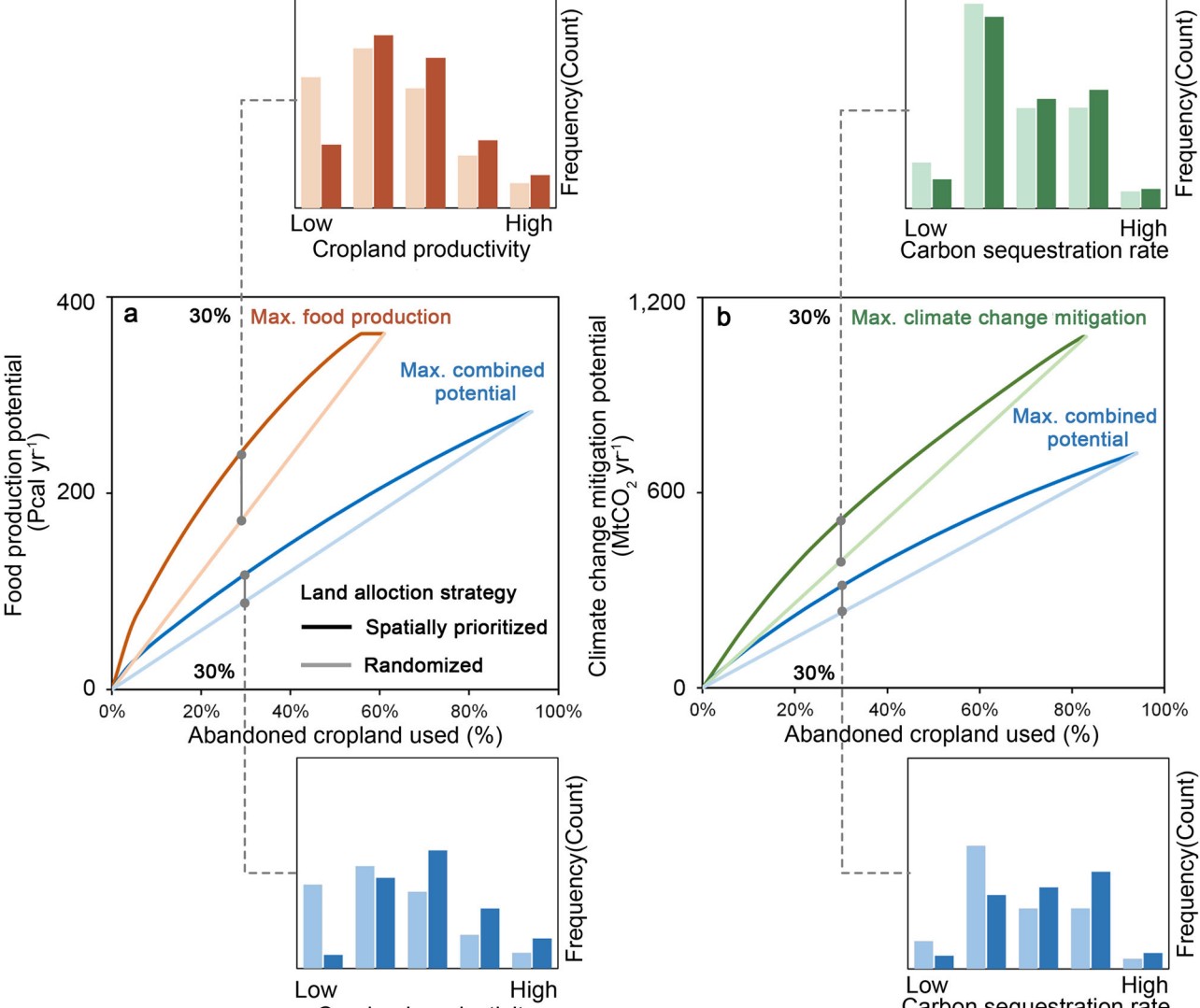

**Fig. 3 | Impact of spatial prioritization on the achievable potential of abandoned cropland.** Food production potential (a) and climate change mitigation potential (b). Histograms in the upper and lower panels present the frequency distributions that summarize cropland productivity and carbon sequestration rates of abandoned cropland that are used (30% of total abandoned cropland). Lines and bars in dark and light colors indicate scenarios employing spatial prioritization strategy and those without (i.e., randomized allocation) in using abandoned cropland, respectively. Results are showcased by three types of representative scenarios.

Eastern Europe, East and Southeast Asia, Central Africa, and South America with Russia, Brazil, and China standing out as top 3 countries (Supplementary Fig. 4).

## Approaches to improve the achievable potential of abandoned cropland

Our estimated food production and climate change mitigation potentials are conservative and slightly underestimate such potentials. There are a few possible approaches to increase the achievable potential of abandoned cropland. Here, we present the impacts of applying well-documented approaches to improve food production potential or climate change mitigation potential, as well as how they would free up land for reforestation or recultivation, respectively (Fig. 4; see Methods and Supplementary Text 3).

The food production potential was estimated by the present-day actual yield of 2010 under rain-fed conditions, representing the most feasible and conservative setting for recultivation. This setting reflects generally less favorable biophysical or socioeconomic conditions in cropland abandonment areas, which would thereby constrain the productivity of recultivated croplands[44,45]. It is also pertinent in the context of limited sustainable water resources for irrigation due to alternative human water usage and the need to protect environmental water flows to sustain freshwater ecosystems, particularly in abandonment hotspots overlapping with sustainable irrigation expansion (e.g., Eastern Europe and Central Africa)[46]. However, these hurdles could be at least partially overcome in the future. First, improving the water supply from a rain-fed to an irrigated condition will increase achievable food production potential by 62%. Second, if cropland yields in areas with identified yield gaps are improved to attainable yields by agricultural intensification and high-level agricultural inputs (i.e., market-oriented agriculture with advanced management[47]), we then predict a 40% increase in food production potential

(Supplementary Figs. 8 & 9). Third, when we estimated food production potential, about 17-34% of food produced was considered either wasted or lost during production through the supply value chain and consumption (see Methods and Supplementary Data 3). If global food waste and loss can be halved, as the target of 12.3 SDGs envisions, the food production potential on recultivated abandoned croplands could be increased by 16%. While all three approaches are hypothetical, if they are all realized, our original estimation of achievable food production potential could be doubled from 363 Pcal yr[-1] to 791 Pcal yr[-1]. Alternatively, if we maintain the food production at our original estimation, applying these approaches would free up an additional 27 Mha of abandoned cropland suitable for reforestation. This would increase opportunities for sequestrating an additional carbon up to 439 MtCO$_2$ yr[-1] (98% more than the original estimation). Future climate change would bring a marginal benefit to crop yields and a 3-12% increase in food production potential, depending on various climate forcing and water supply conditions (Supplementary Table 3). At last, by optimizing the spatial allocation of the most productive crops, the achievable food production potential could be improved by 74% (Supplementary Fig. 6 and Supplementary Table 2).

When modeling climate change mitigation, we based our estimate of the carbon sequestration rates on natural forest regeneration on avoiding adverse impacts on local biodiversity[24]. However, the carbon sequestration rates of natural forest regeneration are lower in non-tropical areas than reforestation approaches with human interventions (e.g., active reforestation). The accumulated aboveground biomass carbon via natural regeneration is reported to take more than 60-80 years to recover to 90% of the old-growth forest level[48,49]. At the same time, abandoned croplands that experience natural regeneration may face intense threats of being recultivated unless policy interventions are implemented[18]. If we actively reforest abandoned cropland (68 Mha, 82%) outside global protected areas defined by IUCN and UNEP,

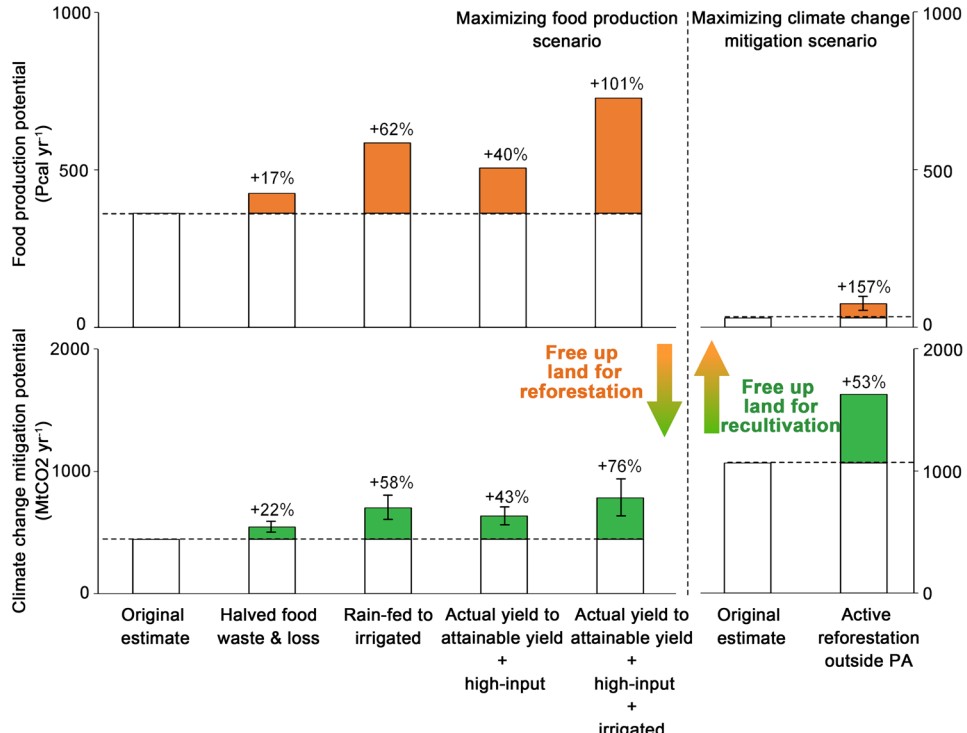

**Fig. 4 | Possible improvements of the achievable potential of abandoned cropland.** Possible approaches to improve achievable food production potential and climate change mitigation potential or to free up land for reforestation (or recultivation) while still achieving the original food production outcomes (or climate change mitigation outcomes). The error bars indicate the mean ± standard deviation of the additional potential obtained from the freed-up abandoned cropland. Results are based on "maximizing food production" scenario and "maximizing climate change mitigation" scenario, with all suitable abandoned cropland used. Attainable yields are the 30-year mean of 2010–2040 under RCP4.5 condition (Supplementary Text 3). PA: protected areas defined by UNEP and IUCN.

our nonspatial estimate predicts an increase of $53 \pm 18\%$ (*mean + std*) on the achievable climate change mitigation potential, with a potential to free up 17 Mha abandoned cropland for recultivation and generate 45 Pcal yr$^{-1}$ food production potential (+153%) (Supplementary Text 3). In our study, we considered that preparing abandoned cropland for recultivation would clear all the historically accumulated aboveground biomass without energy recovery and thereby lead to carbon emission, offsetting the climate change mitigation potential from abandoned cropland. However, it has been reported that the biomass from land clearing can be used for biofuel production. This approach can not only offset the emissions from land clearing but the life cycle emissions of crop production that we did not account for due to the lack of crop-specific data[23]. Agricultural nature-based climate solutions could further create additional climate benefits on abandoned cropland for recultivation. For example, "Trees in croplands" solution, "Conservation agriculture" solution, and "Improved rice cultivation" solution can offer 1.1 MgCO$_2$ yr$^{-1}$ ha$^{-1}$, 1.4 MgCO$_2$ yr$^{-1}$ ha$^{-1}$, and 1.6 MgCO$_2$e yr$^{-1}$ ha$^{-1}$, respectively, compared with the global average carbon sequestration rates of regrowing forest on suitable abandoned cropland (11.5 MgCO$_2$ yr$^{-1}$ ha$^{-1}$)[12].

## Discussion

Our study shows that abandoned cropland can be a promising avenue for unlocking additional land resources in a world where land scarcity remains a major hindrance to sustainable development. If recultivated and/or reforested strategically, abandoned cropland can make a considerable contribution to meeting food security and climate goals. In addition, reforesting abandoned cropland via natural regrowth could bring an array of co-benefits, such as enhancing biodiversity conservation, water filtration, air filtration, and soil quality[12,26,27,50]. Recultivating abandoned cropland would help to offset the geographical displacement of cropland expansion and the subsequent environmental impacts, such as deforestation[51] and wetland loss[52]. Abandoned croplands that area available for reforestation (83 Mha) and recultivation (61 Mha) also show substantial potential to offset future land-use demand, e.g., up to 372 Mha of new cropland expansion and 145-250 Mha of net forest growth (SSPs-26 scenarios by 2050), and ease the intensifying land-use competition[6].

In addition to recultivation and reforestation, abandoned cropland can be used for other purposes, such as growing bioenergy crops[39], grassland restoration[53], and rewilding[29,54]. For example, bioenergy crops have been expected to play an important role in reaching the 2 °C climate goal, with a projected sharp increase of 1.0 to 3.0 Gt dry matter per year from 2020 to 2050 (SSP2-2.6 scenario)[55]. Delays in bioenergy crop deployment would even threaten food security as global warming causes feedback loops with associated decreasing crop yields[56]. Thus, abandoned cropland can complement such rising demand for bioenergy crops. However, there is also debate on bioenergy crops due to their low land-use efficiency, which might exacerbate land competition and lead to displaced cropland expansion and carbon emission[57]. Further scrutiny into these alternative uses of abandoned cropland is needed for the best use of abandoned cropland.

It should be noted that various factors may hamper endeavors to access and utilize abandoned cropland, thus reducing its potential. For example, armed conflicts or regional economic rivalries can have significant impacts on international trade openness, labor availability, market access, and agricultural productivity, which consequently reduce the suitability of abandoned cropland for recultivation and reforestation and/or its potential[16,34,58]. Besides, the ongoing shift towards more meat-intensive diets may limit the potential of abandoned cropland for food production[59], as livestock-based production requires significantly more land than cereal-based production to generate the same amount of calories[14]. In our study, we assumed that all selected crops were consumed directly by humans rather than as animal feed. However, a recent study predicts that 34% of crop production in China will be used for feeding livestock[51]. At last, securing the permanence of food production potential and climate change mitigation potential needs policies and incentives across sectors. Without these efforts, recultivation and reforestation would become ephemeral activities and would consequently reduce the achievable potential of abandoned cropland[18].

While these is a considerable amount of food production and climate change potential achievable from abandoned cropland, further efforts are needed to transform these potentials into grounded benefits. For example, in addition to the increased food provision by recultivating abandoned cropland, it is imperative to improve the accessibility, affordability and effective utilization of food products so that the global food deficit can be addressed[60]. Additional information, such as spatial and demographic data about food insecurity, can guide actions directing limited food production to do the greatest good[61]. Similarly, although the climate change potential obtained from abandoned cropland can offer an important pathway for meeting emissions reduction targets that countries have committed in their NDCs under the Paris Agreement[62], globally collaborative efforts are required to build and scale up a trustworthy carbon market. This is essential to ensure the carbon offset obtained from reforesting abandoned cropland can be coordinated across sectors and countries.

Understanding the trade-offs and spatial priorities in using abandoned cropland is crucial for making land management decisions. Our study demonstrates the plausible outcomes across various trade-off options of abandoned cropland and approaches to maximize the synergies. We also offer broad spatial guidance on where decision-makers should prioritize their land management efforts. These insights are essential to support global and regional initiatives aimed at sustainable land stewardship, such as the Bonn Challenge[63], the UN Decade on Ecosystem Restoration[64], SDG 15[65], Initiative 20-by-20 of COP 20[66], and China's green for grain project[67].

Nevertheless, our global-scale findings have a few limitations. Harnessing and reaching a globally optimized utilization of abandoned cropland will require overcoming the hurdles in global coordination and policy differences. Besides, global maps have been receiving rising critics and debates, particularly on the mismatch in the scale that maps are generated (global scale) and land-use management policies are implemented (usually national or local scales)[68,69]. Even so, these limitations do not undermine the utility of our global-scale findings but rather recognize the importance of across-scale information. Global-scale findings can provide a broader context for local decisions on abandoned cropland management and are essential in such an increasingly interconnected world with globalized decision-making[70]. We emphasize the necessity of incorporating context-specific information at the local scale with our global scale findings to ensure that knowledge on an optimized and synergetic use of abandoned cropland can be transferred into grounded actions that can also benefit local communities. Context-specific information at the local scale includes but is not limited to local land-use policy, land tenure, local land-use policies, culture, cost, and labor availability[71]. Taking local land-use policy, for instance, if the abandoned cropland is already assigned for oil palm plantation with a concession from local government—commonly steered land-use conversion in pantropical areas—recultivation and reforestation would no longer be allowed. Our further calculation shows that, in this case, about 0.66 Mha of abandoned cropland in our study would become unavailable for both purposes (Supplementary Text 4). However, these context-specific data are not always spatially explicit, sometimes even not quantitative. For instance, most currently available maps of global recultivation and reforestation cost only present spatial variation at the country scale, with pixels within a country exhibiting the same or similar cost values[15,28,72]. This hinders the integration of costs, as well as other socioeconomic factors, into the trade-off analysis (see a detailed example in Supplementary

Text 5). Closing this gap would require more open-access data to be made available via field surveys, citizen science, and volunteered geographic information data collection programs[73]. Moreover, to implement local recultivation or reforestation projects on abandoned cropland, it is essential to secure the rights of rural and indigenous people and leverage their knowledge to foster success, enhance resilience, and avoid adverse social and ethical outcomes[72,74]. At last, implementing recultivation and reforestation should also avoid leading to adverse impacts on regional climate. Large-scale reforestation in abandonment hotspots may cause regional water yield insecurity due to relatively higher evapotranspiration than precipitation, particularly in water-stressed or semi-arid regions, such as Northern China, Middle America, East Africa, and Eastern Europe[75]. Reforestation effects should also avoid causing albedo effects (e.g., usually occur in boreal forest expansion), which can lead to net biophysical warming[76,77].

Our study demonstrates that unlocking the potential of abandoned cropland offers substantial benefits for food production and climate change mitigation, while synergies of these benefits open up new opportunities for meeting the Paris Agreement's climate goals, SDG 2 (zero hunger), and SDG 15 (restoring degraded land and increasing afforestation). Our simulated outcomes and trade-off analysis can be translated into valuable insights for policymakers and land-use planners to guide global and regional initiatives. Our analysis also highlights key levers for maximizing the potential benefits of abandoned cropland, including employing spatial prioritization strategies and ensuring a globally coherent effort.

## Methods
We analyzed the potential and trade-offs between recultivation and reforestation of using global abandoned cropland to support food production and climate change mitigation in the 30-year period from 2020 to 2050, with the help of multi-source geospatial datasets, machine learning model, and geospatial and scenario simulation approaches. The following sections describe the datasets and methods used in this study.

### Mapping abandoned cropland
We utilized the ESA-CCI product to identify and map global abandoned cropland. ESA-CCI is a land cover map dataset at a 300-m resolution (10 arc-second) and on a yearly basis from 1992 to 2020[78,79]. This dataset has been widely used for characterizing long-term changes in the Earth's surface, including identifying land cover transitions, analyzing cropland dynamics and mapping cropland abandonment[39,78,80–82]. Following the correspondence IPCC land categorization and the ESA-CCI land cover classification system, there are six land cover classes in ESA-CCI maps representing different cropland types[79]. These cropland classes exhibit global user's and producer's accuracies of 81% and 92% (median), with an overall accuracy of 83%. The classification accuracies of cropland classes are among the highest of all classes in the ESA-CCI product.

We used a relatively conservative strategy to identify abandoned cropland with the annual cropland maps aggregated from all six cropland classes of ESA-CCI. FAO's definition of abandoned cropland was adopted - cropland that has not been used for at least five years; that is, a pixel was first cropland and then converted to non-cropland classes for at least five consecutive years (excluding built-up areas)[39,83]. Three additional operations were implemented to further refine abandoned cropland mapping: (1) a five-year moving window temporal filter was applied to annual cropland maps. If one pixel has a different land cover class, two years before and after will be assigned to the same land cover class. ESA-CCI has already implemented a consistency rule that each change has to be confirmed over more than two successive years in the classification time series. Our temporal filter can further remove unlikely land-cover changes and to improve temporal consistency in cropland maps[18,84], particularly from 1992 to 1999,

which has been reported with a low reliability[79]. It helps to eliminate single-year misclassification in our subsequent abandoned cropland maps and recultivation maps. (2) only historically stable cropland was considered (a pixel was cropland during 1992-1997); (3) cropland converted to settlement and wetland, and abandoned cropland once recultivated (abandoned cropland converted back to cropland for at least one year) were excluded. In this way, we could avoid over-estimating pixels under short-term fallow as abandoned cropland and exclude ephemerally abandoned cropland, e.g., recultivated after a few years' abandonment[18], thereby presenting a relatively reliable estimate of abandoned cropland extent. It should be further noted that we focused on abandoned cropland exclusively because abandoned pasture is challenging to discern from satellite images against natural grassland[18].

Despite the wide use of ESA-CCI in land change studies, ESA-CCI product is not designed for change detection. Mapping cropland abandonment with the ESA-CCI dataset has a few limitations, such as mixed pixel issues (i.e., multiple land-use classes in a coarse resolution pixel), and errors propagated and aggregated from misclassified pixels in each year[81,85,86]. The coarse resolution of ESA-CCI may limit its utility of monitoring cropland abandonment in the regions where agricultural fields are small, such as Africa[87]. Future endeavors to map global cropland abandonment with datasets in higher spatial resolutions (e.g., Landsat and Sentinel-2) could help to address these limitations[87,88]. Here, we used a sampling-based approach to assess the accuracy of our identified abandoned cropland map and to derive error-adjust area estimate. We followed ref. 89 to determine the validation sample size and adopted the disproportionate stratified sampling strategy to validate our abandoned cropland map. This strategy is designed and suitable for assessing the classification accuracy of small classes, such as abandoned cropland[87,89]. For abandoned cropland and nonabandoned cropland classes, we randomly selected 828-pixel samples for each class (1,656 pixels in total; see sample locations in Supplementary Text 6). We visually interpreted these reference samples with the help of the very high resolution (VHS) historical images from Google Earth, and cloud-free Landsat images during the crop growing period from Google Earth Engine[31,32]. Time series spectral indices (e.g., NDVI and Tasseled cap-based greenness) derived from Google Earth Engine also aided validation sample collection[87]. If neither VHS historical imagery nor cloud-free Landsat image is available for a selected sample, this sample would be discarded and a new sample of the same class would be selected. In the case of mixed pixels, we followed the class definition – cropland coverage>50% – based on "Correspondence between the IPCC land categories for change detection and ESA-CCI classes"[79]. To account for possible sampling bias, we reported the overall accuracy, producer's accuracy, user's accuracy, and F1 score of our abandoned cropland map at the global scale and continental scale (Supplementary Tables 5 and 6). Based on the overall accuracy and accuracy assessment protocol[30], we calculated the 95% confidence intervals of the total abandoned cropland extent (101 ± 35 Mha). We acknowledge that removing highly mixed samples and the ones in cloud-frequent regions may have inflated our accuracy estimate, such as in Africa where fields are very small (Supplementary Table 6). We also qualitatively compared the spatial patterns of abandoned cropland identified in our study with previous studies (n = 42; Supplementary Data 1). Note that periods, resolutions, land classification methods, differences in land-use class definition, and abandonment criteria vary between studies, so a quantitative comparison is not practical for most studies.

### Modelling suitability and potential
We considered the two most viable purposes of using abandoned cropland—recultivation for food production and reforestation via natural regrowth for climate change mitigation. Their feasibilities have been confirmed by studies and practices in local and regional

cases[22,31,90,91]. We estimated the suitability and potential of each purpose separately.

We used a machine learning-based approach to estimate the suitability of abandoned cropland for recultivation, trained using driving factors of recultivation and abandoned cropland pixels that were historically recultivated. The potential drivers of recultivation covered biophysical and socioeconomic aspects, including integrated agroecological suitability[47], market accessibility[31], travel time to settlement[92], population density[31], adjacent cropland density[51,93], distance to stable cropland[26], abandoned cropland density[26], INFORM Risk Index of natural and human-induced disasters[94], and annual economic loss due to natural hazards[95] (Supplementary Text 7). Recultivated abandoned cropland was identified as the cropland pixel that was once abandoned and then converted back to cropland at least once during our study period[18]. Based on our analysis with the ESA-CCI dataset, about 2.8 Mha of cropland was once abandoned and then recultivated from 1992 to 2020 (Supplementary Fig. 5). Our identified recultivation map is different from a previous study with 11 study sites[18], which could be ascribed to differences in the definition of cropland abandonment, spatial resolution of input data and study areas. Then, a maximum entropy model (MaxEnt software v3.4.4) was trained using normalized driving factors and historically recultivated pixels[96], and then optimized by factors adjustments and model parameter tuning[97]. We used the fine-tuned MaxEnt model to estimate the abandoned cropland prone to recultivation (recultivatability∈[0,1]). The recultivatable abandoned cropland was determined by a recultivatability larger than 0.2, while pixels with recultivatability values lower than 0.2 were considered as not likely to be suitable for recultivation due to substantial biophysical, agricultural and socioeconomic barriers. At last, protected areas, jointly defined by UNEP and IUCN, were masked out to avoid adverse ecological impact[98]. Model performance and sensitivity analyses are presented in Supplementary Text 8.

We selected 15 major food crops for recultivation based on previous literature[99] and data availability in the Global Agro-Ecological Zoning database (GAEZ v4), a modelling framework co-produced by FAO and IIASA[47]. The GAEZ has been one of the most widely used datasets for providing global-scale estimates of crop-related information, including harvest areas and crop yields[100,101]. GAEZ model and its outputs have been validated by (1) comparisons against other modeling outputs[102,103], (2) comparisons against statistical records[104], and (3) comparisons against crop yields map estimated by bottom-up approaches (i.e., models based on local observations), such as GYGA dataset[105,106] (Supplementary Text 3). The selected 15 crops included barley, cassava, groundnut, maize, millet, oil palm, potato, rapeseed, rice, sorghum, soybean, sugarbeet, sugarcane, sunflower, and wheat. We assumed the cropland productivity of abandoned cropland to be recultivated was the same as the productivity of current active cropland within the same 5 arc-min pixel[91]. The integrated cropland productivity of each pixel was estimated by the area-weighted average of the 15 selected crops based on their present-day harvested areas (Supplementary Fig. 6) and present-day actual yield downscaled from the national annual average of 2009–2011 agricultural statistics under a rain-fed condition[47,99]. It should be noted that 2010 and 2011 are relatively wetter in a 30-year context, and thereby would make this actual yield data higher than the 30-year average condition[107]. Since cropland abandonment areas are subject to less favorable biophysical, climatic, or socioeconomic conditions, it would be challenging for these areas to access the irrigation system or overcome these constraints to close the yield gap[44,45]. We thereby used these conservative but most feasible settings to estimate the achievable food production potential from recultivation in the next 30 years. Nevertheless, we also provided estimations with improved water supply (i.e., irrigation), crop yields, and crop allocations, as well as estimations reflecting the impacts of future climate change (see Methods-Analysis section and Supplementary Text 3). Furthermore, for each crop, we converted its

actual yield from harvest weight yield (kg yr$^{-1}$ ha$^{-1}$) to dry-weight-based calorie yield (kcal yr$^{-1}$ ha$^{-1}$), using conversion factors in GAEZ document[47] and FAO food balance sheets[108]. We also converted integrated food productivity from harvest-level to consumer-level by deducting food waste and loss during production, supply chain and consumption. We adopted the food loss percentage from FAO's food loss index[109] and set a uniform food waste percentage of 17% according to the latest UNEP report[110] (Supplementary Data 3).

For pixel $i$, the integrated productivity from 15 selected crops ($j \in$ [1,15]) is calculated by Eq. (1):

$$Productivity_i = \sum_{j}^{15} yield_j \times \frac{harvested\_area_{i,j}}{\sum harvested\_area_i} \times (1 - food\_loss_i) \times (1 - food\_waste) \tag{1}$$

We identified abandoned cropland suitable for forest natural regrowth with potential natural vegetation (PNV) map[40]. The PNV map presents areas suitable to return to native forest cover that preceded human disturbance based on biophysical, climatic, and lithological conditions. Overlapping areas between 'forest' and 'woodland' classes in the PNV map and our abandoned cropland map allowed us to identify abandoned cropland suitable for reforestation via natural growth[111].

We quantified the annual net climate change mitigation potential in the next 30 years by considering the carbon sequestration achievable from allowing forest natural regrowth on abandoned cropland and the emissions from clearing historically accumulated aboveground biomass carbon for recultivation. First, we utilized the latest map of carbon sequestration rates of forest natural regeneration to calculate annual climate change mitigation potential for the next 30 years[43]. The aboveground carbon sequestration rate map is based on 257 historical studies and 13,112 georeferenced measurements and an ensembled machine learning model. The belowground carbon sequestration rate is estimated post hoc using the IPCC default root-to-shoot ratio[112]. The spatial variation of carbon sequestration rates across different climate zones, land-use types, forest types, and other factors are better presented, which is essential for identifying high carbon sequestration rate areas for prioritized actions. Aboveground and belowground biomass carbon was considered, but not soil organic carbon. It was because the soil carbon accumulation rates after natural regrowth were found negligible or negative, even contrasting across studies[43,90,113]. Unlike changes in aboveground and belowground biomass carbon, soil carbon change during forest natural regrowth is understudied and short of measurements. While this carbon sequestration rate map is one of the best available data, there are factors that the mapping failed to account for, such as differences in forest stages and local water supply conditions. We thereby present the uncertainty of the carbon sequestration map and its impact on our climate change mitigation estimation using the corresponding error rate layer (i.e., one standard deviation of spatial variability across 100 trained machine-learning models) (Supplementary Text 9).

Second, we estimated the emission from clearing the historically accumulated aboveground biomass carbon when preparing the abandoned cropland for recultivation. We assumed that all the accumulated aboveground biomass carbon would be immediately oxidized and lead to one-time emission[23]. The current land cover condition was obtained from our ESA-CCI land cover map in 2020. We found 65% of areas suitable for reforestation were in the process of natural regeneration, while the remaining areas were grass (14%), shrub (10%) and bareland (11%). For areas under natural regeneration, the accumulated biomass carbon was directly estimated by the map of Cook-Patton, et al.[43]. For abandoned cropland that was currently grass, shrub and bareland, the accumulated biomass carbon was estimated by the carbon stock map from Spawn, et al.[114]. The breakdowns of net climate

change mitigation potential are presented in Supplementary Text 2. In addition to land clearing, recultivating abandoned cropland may lead to other emissions during the crop production cycle. For example, wheat production in Western Australia was estimated to incur a life cycle emission of 308 kgCO$_2$e t$^{-1}$ during prefarm, on-farm, and post-farm stages[115]. However, we did not include life cycle emission of recultivation in our analysis mainly because of the lack of spatially explicit life cycle emission data for all 15 crops considered in our study.

In this way, we obtained maps of suitable abandoned cropland for recultivation and reforestation. We further categorized them into four suitability types, including areas only suitable for either recultivation or reforestation, areas suitable for both, and areas suitable for neither purpose (Fig. 1b). We also obtained maps of food production potential (integrated cropland productivity) and climate change mitigation potential (carbon sequestration rates) in suitable abandoned cropland (Fig. 1c, d and Supplementary Fig. 1).

### Evaluating trade-offs and synergies between recultivation and reforestation

Based on the modeled suitability and potential, we simulated scenarios to evaluate trade-offs between recultivation and reforestation. These simulated scenarios present the outcomes of different options in utilizing abandoned cropland. Each simulation was determined by the amount of abandoned cropland to be used in total and for either purpose, the locations of abandoned cropland to be allocated for each purpose, and the way of allocation (randomized or spatially prioritized). For simulations applying a spatially prioritized land allocation strategy, pixels with higher integrated cropland productivities or carbon sequestration rates were allocated first. It should be further noted that each 300-m abandoned cropland pixel was only allocated for a single purpose, either recultivation or reforestation. This scenario analysis serves to contrast different options in using abandoned cropland and determine how synergies of food production and climate change mitigation could be maximized. A detailed explanation of the simulation processes is presented in Supplementary Text 10. The outcomes of all scenarios are visualized in Supplementary Fig. 2.

### Analysis

To assist comparison of the simulated outcomes across scenarios, we calculated the following metrics in addition to food production potential and climate change mitigation potential: (1) the percentage of achieved potential relevant to the maximum achievable food production potential (363 Pcal yr$^{-1}$; $A$) and climate change mitigation potential (1080 MtCO$_2$ yr$^{-1}$; $B$); (2) the achieved combined potential in percentage ($A + B$), which serves as a hypothetical indicator to demonstrate the integrated outcome of recultivation and reforestation of each scenario.

To better illustrate and compare the simulated outcomes across scenarios, we showcased four types of hypothetical but representative scenarios: "maximizing food production" scenario, "maximizing climate change mitigation" scenario, "equal allocation" scenario", and "maximizing combined potential" scenario. For "maximizing combined potential" scenario, whether a pixel was allocated for recultivation or reforestation was determined by its comparative advantage over cropland productivities and carbon sequestration rates (see detailed scenario narratives in Supplementary Table 9).

We assessed the impact of applying a spatially prioritized allocation strategy on the resulting potential of abandoned cropland by comparing outcomes simulated with and without spatial prioritization across different percentages of abandoned cropland that are used (Supplementary Text 11). The frequency distributions of cropland productivity and carbon sequestration rates were also compared across scenarios with and without applying spatial prioritization. These comparisons allowed us to evaluate the additional potential that can be unlocked through spatial prioritization and to demonstrate the

importance of strategic land management in maximizing the potential of abandoned cropland.

Our estimates of the food production potential and climate change mitigation potential are conservative. There are a few possible approaches to improve the achievable potentials and free up land for other use. We considered the following possible approaches and quantitatively assessed their benefits in improving the achievable potential of abandoned croplands. For recultivation, we considered the impacts of improvements in water supply conditions from rain-fed to irrigated, crop yields from actual yields to attainable yields for areas with identified yield gaps (Supplementary Figs. 8 and 9), crop allocation method from depending on present-day harvest area to maximizing the integrated crop productivity of each pixel, halving the food waste and loss, as well as impacts of future climate change under different climate forcing (RCP45[116] and RCP85[117]). The future climate forcing was the ensemble mean of five climate models GFDL-ESM2M, HadGEM2-ES, IPSL-CM5A-LR, MIROC-ESM-CHEM, and NorESM1-M, obtained from the bias-corrected Intersectoral Impact Model Intercomparison Project (ISI-MIP)[118]. For reforestation, we considered the potential improvement in climate change mitigation potential if we actively reforest suitable abandoned cropland instead of via natural regrowth (see Supplementary Text 3 and Supplementary Data 4). We only considered active reforestation on abandoned cropland outside the global protected area to avoid irreversible and adverse influences on biodiversity while reforesting the remaining abandoned cropland via natural regrowth (Supplementary Fig. 11). Details of these approaches and how we estimated the potential improvements are presented in Supplementary Text 3.

### Reporting summary

Further information on research design is available in the Nature Portfolio Reporting Summary linked to this article.

## Data availability

All input data used in this study are publicly available from the corresponding references. The administrative boundaries were obtained from World Bank Official Boundaries. The gridded data of aboveground and belowground carbon accumulation rates of natural regrowth were provided by Susan Cook-Patton (ref. 43). All data generated for this study are available via Zenodo data repository (https://doi.org/10.5281/zenodo.8010675). Source data are provided with this paper.

## Code availability

Custom codes developed for this study are available via Zenodo data repository (https://doi.org/10.5281/zenodo.8010675).

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

## Acknowledgements

L.P.K. is supported by the National Research Foundation, Singapore, under its NRF Returning Singaporean Scientists Scheme (NRF-RSS2019-007). Q.Z. is supported by the Strategic Hiring Scheme Fund of the Hong Kong Polytechnic University (P0044791). Research contributes to the Global Land Programme's "Agricultural Land Abandonment as a Global Land-Use Change Phenomenon" working group. We thank Mr. Ning Zhu and Ms. Hanci Liang for their help and suggestions on visualization.

## Author contributions

Q.Z., T.H., and L.P.K. designed this study. Q.Z. developed the code, analyzed the results, and wrote the initial manuscript. Q.Z., T.H. and A.V.P. collected the data. Q.Z., T.H., A.V.P., Y.Z., H.Y., and L.P.K. contributed to writing, editing, and commenting the manuscript.

## Competing interests

The authors declare no competing interests.
