## [Peer Review File · Nature Communications]

The neglected role of abandoned cropland in supporting both food security and climate change mitigationEditorial Note: Parts of this Peer Review File have been redacted as indicated to remove third-party material where no permission to publish could be obtained.

Reviewers' Comments:

Reviewer #1:
Remarks to the Author:
Review Nat. Com

Thank you, authors, for this study. The study aims to fill an important gap in the literature by comparing the global benefits of recultivation of abandoned cropland for food production and alternative land-based climate change mitigation through active afforestation at high spatial resolution. I found the manuscript novel and well-written and enjoyed reading it. The topic should be of broad interest. However, I also found that the present manuscript falls short of addressing this important topic in a convincing way. I have several major methodological concerns that should be solved, such as related to the quantification of active reforestation rates, lack of soil carbon consideration, and unclear time horizon. The analysis does not currently reach its untapped potential. I hope my suggestions can help the authors strengthen their study further.

Main strengths:

1. The manuscript critically advances the literature by spatially quantifying recultivation potentials for food production and achievable climate change mitigation of active reforestation at a global scale.
2. It proposes prioritization strategies for the management of abandoned croplands that may help achieve global sustainability goals. This is a much-needed contribution and provides compelling insights.
3. A wide range of models/data sources are integrated to address the research questions at hand in a new manner.
4. The paper is overall well-written.

Main comments:

1. Climate change mitigation. It should be clearer from the beginning that the "mitigation" scope of this paper is active reforestation as an alternative to natural regrowth. Reading the title and the abstract now, there is no way to identify that you are addressing active reforestation, and not one of the many other land-based mitigation measures available as future trajectories for abandoned cropland. This should be clarified both in the abstract and title.
2. Carbon fluxes over time. Arguably, what is here called "baseline carbon sequestration conditions" (e.g., natural regrowth) is already an ongoing process that removes carbon dioxide from the atmosphere over time relative to the previous land use (cropland). The authors point out that the "additionality" relative to baseline conditions is the only mitigation that is tradable as carbon credit. Therefore, the difference between active reforestation and natural regrowth is quantified as mitigation. This thus raises the question, what is in fact the business as usual?

Your results indicate minor recultivation activity (Ext. Data Fig. 6). However, according to Crawford et al. (2022) that considered satellite observations from Landsat at 30m resolution (85% overall accuracy claimed), cropland abandonment is highly ephemeral and more than half of historical cropland abandonment in abandonment hotspots have typically been recultivated within 30 years. So, who has the more reliable data? There is clearly a disagreement there. If the findings of Crawford et al. are correct, does it still make sense to rely on continued natural regrowth as a baseline and only show the difference? Or is avoiding recultivation in fact already an intervention requiring active policy measures?

I suggest a change in thinking towards including negative emissions over time of continued natural regrowth in some of your results. It would improve the overall comprehensibility to show average yearly carbon fluxes over a given period (for example, 30 years based on Cook-Patton data limitations) simultaneously for both natural regrowth, active reforestation, and recultivation. The additionality of carbon dioxide removal contribution from active reforestation will be clearly visible.

See for example, Field et al. Figs 3 and 4 for an example comparison of continued regrowth/bioenergy/BECCS.

Crawford, C. L., Yin, H., Radeloff, V. C., & Wilcove, D. S. (2022). Rural land abandonment is too ephemeral to provide major benefits for biodiversity and climate. *Science Advances*, 8(21), eabm8999. <https://www.science.org/doi/10.1126/sciadv.abm8999>

Field, J.L., Richard, T.L., Smithwick, E.A.H., Cai, H., Laser, M.S., LeBauer, D.S., Long, S. P., Paustian, K., Qin, Z., Sheehan, J.J., Smith, P., Wang, M.Q., Lynd, L.R., 2020. Robust paths to net greenhouse gas mitigation and negative emissions via advanced biofuels. *Proc. Natl. Acad. Sci. USA* 117, 21968–21977. <https://doi.org/10.1073/pnas.1920877117>

3. Climate impacts of recultivation. It is inconsistent not to quantify the climate impacts of recultivation and on-farm activities when designing prioritization strategies between climate change mitigation and recultivation for food production. Emissions related to land use change (i.e., the clearance of historically accumulated aboveground carbon) are quantifiable by the datasets you already have at hand. Furthermore, life cycle emissions related to on-farm activities such as usage of machinery, fertilizer, pesticides etc. and supply chain emissions should be accounted for. The lost mitigation of foregone natural regrowth should also be shown.

4. Active reforestation carbon accumulation (key point, hence long comment!). The Cook-Patton machine-learning dataset is a robust machine-learning dataset providing natural regrowth rates at high resolution and likely the best one available. This study makes use of a global conversion factor to spatially quantify active reforestation by multiplying grid-specific natural regrowth rates for all pixels with the conversion factor. This conversion factor is based on relatively few studies (n=27, Extended Data File 2), and the spatial locations of observations are heavily centered in a few countries (mainly in the tropics). Thus, applying this global factor to a gridded abandoned cropland map with a different spatial distribution of available land means you are extrapolating the average global factor into locations that are not covered by your gathered data. Does this even make sense? This degrades the initial robustness coming from the Cook-Patton dataset that is based on a much larger set of observations and more sophisticated modelling techniques (although data gaps are also present there). The conversion factor variation is also large, even within countries. Furthermore, it is not shown if factor variation arises due to chosen tree species, different on-site management, or different local climatic conditions. I don't find it sufficient to only explore uncertainty through the standard deviation of variation arising from this limited information from 27 studies, mainly due to the geospatial variation of abandoned cropland intensity. Also, the major uncertainty shown undermines the significance of the study. I'd suggest the authors to consider the following actions:

i) Compare the mapped active reforestation carbon accumulation rates with output from other models. Preferably, this should include a spatial component and cover either the geographical regions, climate zones, biomes, or plant functional types that are dominant in the areas with most intense cropland abandonment (e.g., Eastern Europe, Brazil, Central Africa, Indonesia, Central America). Models that could be compared against include LPJmL, CLM, or G4M ++ (and yes, these models also have their own uncertainties, prediction errors, and limitations). On the top of my head, I can suggest checking Doelman et al. (2020) that has some regional/country insights and Braahekke et al. (2019). There should be much more literature out there.

ii) Gather more data to support their modelling effort. Show how the conversion factor correlates with climatic conditions (e.g., temperature/precipitation) and active/passive management, and/or how it varies across broadleaf/evergreen forests, geographical regions, biomes, or climatic zones, and assessed time horizon. Use these insights to improve their model.

iii) Gather observational data on aboveground carbon accumulation from active reforestation projects for the sole purpose of validation. Compare predicted carbon accumulation rates from the active

reforestation map with observations at a locational basis and produce some statistical insights.

iv) Instead of relying on the combination of a machine-learning dataset and a simplified conversion factor, run a process-based land surface model to quantify active reforestation carbon accumulation.

v) Drop the conversion factor altogether in the spatial analysis, and instead rely directly on the Cook-Patton dataset. This might mean changing the research question towards finding priority areas for continued natural regrowth instead of for active reforestation. This still provides a novel contribution. The global average conversion factor can still be used to show an aggregated global mitigation estimate of intervening through active reforestation across scenarios, but then without presenting spatial results and whilst clearly explaining the underlying limitations.

There are several gridded masks and shapefiles openly available that could help you filter out country-level, climate zone specific, or biome specific data (you may know of them already). For example, CIESIN (2018), Beck et al. (2018), and Olson et al. (2001) cited below.

Beck, H., Zimmermann, N., McVicar, T. et al. Present and future Köppen-Geiger climate classification maps at 1-km resolution. *Sci Data* 5, 180214 (2018). <https://doi.org/10.1038/sdata.2018.214>

Braakhekke, M. C., Doelman, J. C., Baas, P., Müller, C., Schaphoff, S., Stehfest, E., & Van Vuuren, D. P. (2019). Modeling forest plantations for carbon uptake with the LPJmL dynamic global vegetation model. *Earth System Dynamics*, 10(4), 617-630. <https://doi.org/10.5194/esd-10-617-2019>

Center for International Earth Science Information Network - CIESIN - Columbia University. 2018. Gridded Population of the World, Version 4 (GPWv4): National Identifier Grid, Revision 11. Palisades, New York: NASA Socioeconomic Data and Applications Center (SEDAC). <https://doi.org/10.7927/H4TD9VDP>

Doelman JC, Stehfest E, van Vuuren DP, et al. Afforestation for climate change mitigation: Potentials, risks and trade-offs. *Glob Change Biol*. 2020;26:1576–1591. <https://doi.org/10.1111/gcb.14887>

Olson, D. M., Dinerstein, E., Wikramanayake, E. D., Burgess, N. D., Powell, G. V. N., Underwood, E. C., D'Amico, J. A., Itoua, I., Strand, H. E., Morrison, J. C., Loucks, C. J., Allnutt, T. F., Ricketts, T. H., Kura, Y., Lamoreux, J. F., Wettengel, W. W., Hedao, P., Kassem, K. R. 2001. Terrestrial ecoregions of the world: a new map of life on Earth. *Bioscience* 51(11):933-938. <https://www.worldwildlife.org/publications/terrestrial-ecoregions-of-the-world>

5. Soil carbon change. There is no consideration or even discussion of soil carbon implications of the different land uses you propose. An attempt to quantify this would be beneficial. Some discussion is a necessity. Cook-Patton et al. shows some belowground carbon accumulation rates, both as global averages and biome-specific (although, large variation even within biomes). Bell et al. is relevant. Please review some literature.

Bell, S. M., Barriocanal, C., Terrer, C., & Rosell-Melé, A. (2020). Management opportunities for soil carbon sequestration following agricultural land abandonment. *Environmental Science & Policy*, 108, 104-111. <https://doi.org/10.1016/j.envsci.2020.03.018>

Cook-Patton, S.C., Leavitt, S.M., Gibbs, D. et al. Mapping carbon accumulation potential from global natural forest regrowth. *Nature* 585, 545–550 (2020). <https://doi.org/10.1038/s41586-020-2686-x>

6. Aboveground carbon stocks on abandoned cropland were not quantified. Having an idea of the historical carbon accumulation on the abandoned cropland is important for a couple of reasons. i) recultivation for food production means there is a need to clear the secondary vegetation. If this biomass is combusted this creates a non-neglectable initial emission pulse that contributes to radiative

forcing and that should be quantified and accounted for. ii) Aboveground carbon may also need to be cleared before tree planting depending on the local conditions.

This has already been done at least twice. First by Crawford et al. (2022) in abandonment hotspots by integrating Landsat and Cook-Patton, then by Næss et al. (2023) for the Nordics by integrating ESA CCI-LC and Cook-Patton. The authors clearly have the data at hand to do the same, and I see no scientific reasons not to consider this as it has been shown to be important.

Especially, the fate of this biomass is also important, whether it is burned at site without energy recovery, combusted with energy recovery that may replace fossil fuels (e.g., wood stoves, or fed to modern bioenergy supply chains to produce biofuel, bioelectricity, or hydrogen), or even used as feedstocks to BECCS. As a minimum, I'd suggest considering that aboveground carbon is combusted at site without energy recovery. The time horizon also matters for the relative importance of clearing aboveground carbon. Longer time horizons mean it gets less important.

Crawford, C. L., Yin, H., Radeloff, V. C., & Wilcove, D. S. (2022). Rural land abandonment is too ephemeral to provide major benefits for biodiversity and climate. *Science Advances*, 8(21), eabm8999. <https://www.science.org/doi/10.1126/sciadv.abm8999>

Næss, J. S., Hu, X., Gvein, M. H., Jordan, C. M., Cavalett, O., Dorber, M., ... & Cherubini, F. (2023). Climate change mitigation potentials of biofuels produced from perennial crops and natural regrowth on abandoned and degraded cropland in Nordic countries. *Journal of Environmental Management*, 325, 116474. <https://doi.org/10.1016/j.jenvman.2022.116474>

7. The considered time horizon is unclear. Results are frequently presented on a per year basis without clarification of considered period. The used datasets limit how far into the future you can claim to assess. The Cook-Patton dataset considers 30 years. GAEZ typically predicts 30-year average food crop yields depending on background climate. Actual historical yields from GAEZ may be based on shorter time horizons (e.g., single years). Hanssen et al. (2020) shows how important chosen time horizon can be when assessing climate change mitigation potentials of negative emission technologies. The importance of time horizon when assessing carbon dioxide removal is also highlighted by Terlouw et al. (2021).

Hanssen, S.V., Daioglou, V., Steinmann, Z.J.N. et al. The climate change mitigation potential of bioenergy with carbon capture and storage. *Nat. Clim. Chang.* 10, 1023–1029 (2020). <https://doi.org/10.1038/s41558-020-0885-y>

Terlouw, T., Bauer, C., Rosa, L., & Mazzotti, M. (2021). Life cycle assessment of carbon dioxide removal technologies: a critical review. *Energy & Environmental Science*, 14(4), 1701-1721. <https://doi.org/10.1039/D0EE03757E>

8. GAEZ crop yields. The methodological description made me confused (lines 478-506). It is written that present day actual rain-fed yields at low agricultural input as 30-year means (1980-2010) are considered. In GAEZ v4, there are three types of yield outputs, i.e., agro-climatic yields, attainable yields, and actual yields. Actual yields are derived through downscaling annual national average yields over 3-year periods (for example, for 2010 based on the years 2009-2011) from agricultural statistics (FAOSTAT). Only the GAEZ modelled agro-climatic yields and attainable yields are representative of 30-year averages, and they thus rely on different background climates (e.g., historical climatic conditions from CRUTS-32, or for future RCPs from Earth System Models). The actual yields are also not modelled based on agricultural input (as claimed in the manuscript), this is only the case for agro-climatic and attainable yields. Please clarify better what was used and ensure that the methodological description aligns with the GAEZ model documentation. If you considered attainable yields, then make sure to describe and cite the correct background climate dataset/model.

<https://www.fao.org/3/cb4744en/cb4744en.pdf>

9. It is claimed that GAEZ “low agricultural input” is considered in the main analyses, but it is never explained what this means, and it needs to be clarified. To quote the GAEZ model documentation:

“Low-level inputs/traditional management

Under the low input, traditional management assumption, the farming system is largely subsistence based and not necessarily market oriented. Production is based on the use of traditional cultivars (if improved cultivars are used, they are treated in the same way as local cultivars), labor intensive techniques, and no application of nutrients, no use of chemicals for pest and disease control and minimum conservation measures.”

Considering yield predictions based on no mechanization, no fertilizer use, no pesticides, etc., will likely produce an underestimation of recultivation potentials in developed countries. It is acknowledged that estimates are conservative (e.g., underestimation). GAEZ has the capability of providing yield predictions for more modern agricultural management conditions. Why not use the model’s capabilities to produce estimates of attainable yield reflecting actual management variation? Considering a spatially differentiated management factor accounting for varying yield gaps in combination with high/low input yields from GAEZ can provide insights. Yield gaps are available from the GAEZ database.

Extended Data fig 5 provides valuable insights into how both improved management and irrigation might increase potentials and help free land for re-forestation. I found this figure very helpful. However, I was left wondering how this affects the spatial prioritization. Maybe you can include a SI figure showing this. Also, please describe in the methods what high inputs means in GAEZ (see GAEZ documentation cited above).

10. Missing insights into crop distribution. 15 food crops are considered based on historical allocation patterns. However, there is no transparency into their spatial deployment, the extent of land allocated to different crops, or what crops provide the dominant contribution to production potentials. Which crops are the important ones? Presenting a map of dominant crop allocation and quantifying these things would help the reader’s understanding. What about the effect of optimizing the crop distribution for maximized calorie production? Would this yield a substantial gain in potential?

11. Lack of validation of crop yields. Have GAEZ predicted attainable food crop yields been validated against observations? If yes as I would expect, it seems appropriate to provide some indication of performance with associated references. Otherwise, providing some validation with comparison of predictions and observations might be necessary.

12. Abandoned cropland validation. While ESA CCI-LC has been used before to map abandoned cropland (e.g., Næss et al. (2021) and Leirpoll et al. (2021)), this study aims to provide for the first time an accuracy assessment of an ESA CCI-LC based abandoned cropland map (80% overall accuracy claimed) instead of relying only on cropland class accuracies and cross-study comparisons. This is an important contribution. However, it is currently not transparent enough to evaluate it properly. The claim should be supported by shown data.

First, I was left wondering why the underlying confusion matrices were not provided in a supplement as for example done in the SI of Lesiv et al. (2018)? It is recommended by Olofsson et al. (2013) to include them.

Second, your sample size also seems relatively small (n=500), whilst for example Lesiv et al (2018) used about 6000-pixel samples in their abandoned cropland validation. I don’t see any justification for why the current sample size is sufficient in this case or discussions regarding sample size. Also, please provide the coordinates and classification of your sample in the extended data as done by Lesiv et al.

Third, it would be beneficial to bring up estimated producer and user accuracies of the abandoned cropland class in the main text for enhanced understanding. Could be compared with previous studies such as Lesiv et al. Overall accuracy seems not enough.

Fourth, did you find any geospatial (i.e., for different regions or climate zones) or temporal variation in abandoned cropland accuracy? Such spatiotemporal insights of map accuracy would be very helpful when assessing the reliability of the abandoned cropland map at a continental, regional, or local level.

Leirpoll, M. E., Næss, J. S., Cavalett, O., Dorber, M., Hu, X., & Cherubini, F. (2021). Optimal combination of bioenergy and solar photovoltaic for renewable energy production on abandoned cropland. *Renewable Energy*, 168, 45-56. <https://doi.org/10.1016/j.renene.2020.11.159>

Lesiv, M., Schepaschenko, D., Moltchanova, E. et al. Spatial distribution of arable and abandoned land across former Soviet Union countries. *Sci Data* 5, 180056 (2018). <https://doi.org/10.1038/sdata.2018.56>

Næss, J.S., Cavalett, O. & Cherubini, F. The land–energy–water nexus of global bioenergy potentials from abandoned cropland. *Nat Sustain* 4, 525–536 (2021). <https://doi.org/10.1038/s41893-020-00680-5>

Olofsson, P., Foody, G. M., Stehman, S. V. & Woodcock, C. E. Making better use of accuracy data in land change studies: Estimating accuracy and area and quantifying uncertainty using stratified estimation. *Remote Sens Environ* 129, 122-131, doi:10.1016/j.rse.2012.10.031 (2013).

13. Currently, there is no context provided as numbers regarding the future land use/cover changes associated with scenarios consistent with the Paris Agreement. It may be useful to present the afforestation extent and carbon accumulation seen in 2C scenarios towards 2050 from the Shared Socio-Economic Pathways (e.g., from Integrated Assessment Models, SSPx-RCP2.6/1.9) in the introduction. Comparing your results with such future mitigation pathways would also help make it easier to understand the potential global contribution of abandoned croplands to achieve mitigation targets.

https://www.ipcc.ch/site/assets/uploads/sites/4/2020/02/SPM_Updated-Jan20.pdf

14. Recultivation for food production might help avoid cropland expansion into natural areas and prevent associated aboveground carbon losses or biodiversity impacts. It would be useful also here to link the land availability found here to SSP projections. Comparing food production potentials to increased food demand projected by IAMs in addition to the food deficits may also help.

15. Alternative land-based mitigation measures (discussion, lines 292-297). The presented argument against recultivating abandoned cropland for bioenergy deployment seems a bit of a cherry pick. Yes, the food vs fuel debate/land use competition is important, but IAMs show that bioenergy expansion is necessary to reach ambitious climate targets. The associated land use changes for bioenergy crops typically matches or even surpasses those for afforestation (see for example, IPCC AR6, IPCC Special Report on Climate Change and Land, or any IAM paper addressing future land use). Xu et al. (2022) shows that delaying bioenergy crop deployment may in fact threaten food security as global warming causes feedback loops with associated decreasing crop yields. Arguably, reforestation and bioenergy/BECCS recultivation should be considered together, and the best use of land will likely vary based on local conditions.

Xu, S., Wang, R., Gasser, T. et al. Delayed use of bioenergy crops might threaten climate and food security. *Nature* 609, 299–306 (2022). <https://doi.org/10.1038/s41586-022-05055-8>

16. A key output from this study is global priority maps. The authors claims assertively that spatially prioritized land management strategies are a necessity to make optimal use of abandoned cropland. There is an unacknowledged ongoing debate on the usefulness of global priority maps. For example, Wyborn & Evans (2021) argues that global priority maps constitute a problematic global knowledge that removes local context. Or as argued by Meyer & Pebesma (2022), global maps create a strong satisfactory feeling that can be too easily abused. This is in sharp contrast to the argument presented here. There is already some good discussion provided, but I believe that bringing your arguments into the context of this debate may help increase the impact of the study.

Wyborn, C., Evans, M.C. Conservation needs to break free from global priority mapping. *Nat Ecol Evol* 5, 1322–1324 (2021). <https://doi.org/10.1038/s41559-021-01540-x>

Meyer, H., Pebesma, E. Machine learning-based global maps of ecological variables and the challenge of assessing them. *Nat Commun* 13, 2208 (2022). <https://doi.org/10.1038/s41467-022-29838-9>

17. Biophysical effects of land use changes also vary based on forest types/climate zones. I suggest to highlight how this may affect the regional climate in different abandonment hotspots in the discussion section. Cook-Patton can point you to some key papers.

Other comments:

Introduction: Some more context from the SSPs on projected land use change would be helpful (show numbers).

Line 76-78: Leirpoll et al. (2021) also used this dataset. Discrepancy should be mainly due to the longer period, as the cropland abandonment rate is very similar.

Leirpoll, M. E., Næss, J. S., Cavalett, O., Dorber, M., Hu, X., & Cherubini, F. (2021). Optimal combination of bioenergy and solar photovoltaic for renewable energy production on abandoned cropland. *Renewable Energy*, 168, 45-56. <https://doi.org/10.1016/j.renene.2020.11.159>

Figure 1. The color scales are a bit hard to read without intermediate x-ticks (same with fig 2). Figure 1b seems like a binary yes/no? If that is correct, please clarify in caption.

Lines 299-315: This is very useful. I'd suggest also pointing out that sustainable water resources for irrigation are limited due to alternative human water usage and the need to protect environmental water flows to sustain freshwater ecosystems, but also that especially the abandonment hotspot in Eastern Europe overlaps with sustainable irrigation expansion opportunities (see Rosa et al. (2020)).

Rosa, L., Chiarelli, D. D., Sangiorgio, M., Beltran-Peña, A. A., Rulli, M. C., D'Odorico, P., & Fung, I. (2020). Potential for sustainable irrigation expansion in a 3 C warmer climate. *Proceedings of the National Academy of Sciences*, 117(47), 29526-29534. <https://doi.org/10.1073/pnas.2017796117>

Lines 399-417: ESA CCI-LC already has a 3-year consistency rule, such that a detected land use change needs to be consistent for three consecutive years before appearing in the dataset (for example., a pixel transitioning from cropland to a non-cropland class first in 1993 needs three consecutive years detected as non-cropland before appearing in the dataset with non-cropland classification in 1995). Not sure if this was considered.

Lines 464-466: You arrive at very different conclusions regarding the historical importance of recultivation than Crawford et al. (2022). Are there any known arguments that may cause this discrepancy (maybe resolution, CCI's 3-year consistency rule (speculating))? Cross-study comparisons and some discussion seems appropriate.

Crawford, C. L., Yin, H., Radeloff, V. C., & Wilcove, D. S. (2022). Rural land abandonment is too

ephemeral to provide major benefits for biodiversity and climate. *Science Advances*, 8(21), eabm8999. <https://www.science.org/doi/10.1126/sciadv.abm8999>

Line 539: There is not any gridded belowground carbon accumulation data provided by Cook-Patton. Please revise.

Lines 557-568: As I read this, it is assumed that tree planting can be done on abandoned cropland without first clearing the historically accumulated aboveground carbon. I think it is highly questionable to generally assume this for areas that have already been abandoned for 20-30 years. It may however be ok for newly abandoned areas or cropland that transitioned to non-forested ecosystems. It is a necessity to consider different scenarios for the fate of standing biomass on abandoned cropland at the time of active reforestation intervention. If it gets burned at site, there is an associated emission pulse involved. The authors should consult the literature to unravel the typical treatment of standing biomass on abandoned cropland and provide an assessment of its importance.

SI lines 235-240: The error ratios provided by Cook-Patton refers to one standard deviation of spatial variability across 100 trained machine-learning models (see Cook-Patton et al. (2020), page 9, or their online dataset description). This should be clarified better. Also, not clear to me if you did some processing to go from one standard deviation of cross-model variability to 95% confidence intervals in Supplementary Figure 5, or if this range is mis-termed as CI.

Quote Cook-Patton et al.: "Because our model is an ensemble of 100 random forest models with each random forest model trained on an independent bootstrapped sample of the training data, we can use the standard deviation of the 100 random forest models' predictions to estimate model uncertainty in each pixel. Therefore, for each pixel we have the model's prediction and standard deviation across the 100 models."

<https://www.nature.com/articles/s41586-020-2686-x>

Possible minor typos:

Line 26: "pace" should be "place".

Thank you again for the very interesting read. I sign this in support of Nature Communication's important push towards transparent peer review.

Best regards,
Jan Sandstad Næss

Reviewer #2:

Remarks to the Author:

Review for: The neglected role of abandoned cropland in supporting both food security and climate change mitigation

Key Results:

This paper provides a new map of global abandoned cropland, and evaluates how the use of that abandoned cropland for either food production or carbon sequestration by active reforestation can help reduce global hunger and help countries meet carbon emissions reductions goals. By estimating the spatial heterogeneity in both crop yields and carbon sequestration potential, the authors are able to further present an optimal spatial pattern of conversion of abandoned cropland into the two categories (food production or reforestation) to achieve the greatest combined benefit. They show that this carefully considered pattern is better than randomly allocating the land. Most importantly, they show that making use of abandoned cropland can substantially contribute to alleviating pressure on global land use.

Validity:

The authors took great care to thoroughly consider the many factors that influence the possibility of re-cultivation and reforestation. These results, along with the conservative estimates of food produced and carbon sequestered appear valid with appropriate levels of uncertainty quantification, underlying data availability, and method explanation provided.

However, I find the conclusions regarding global food deficit reductions are not fully robust due to the lack of consideration of the global agricultural economy. In many nations, people lack sufficient calories because they cannot afford food; this issue is related but not exactly the same as insufficient food supply. I am not suggesting the authors try to include economics in their (already very detailed) analysis. Rather, I would like to see the conclusions stop at the level of calories produced or number of people who could be fed by those calories. The discussion section or supplementary text should then add a brief section on the difference between increasing food production and actually making food accessible to low-income populations.

The main result is a global optimization for how to use abandoned cropland. However, it is highly unlikely that all countries in the world will coordinate in a way that would enable such an optimization. The authors should address this limit in the Discussion section, as the result is valid in a hypothetical way, but not in a real-world policy recommendation way.

Significance:

These findings are significant to the field of global agricultural production, land use categorization, and the land-energy-food nexus. Perhaps of greatest significance and value to the research and policy analysis community is the data itself, specifically the geospatial data of abandoned cropland, and potential yields and potential carbon sequestration on each abandoned cropland pixel. For this reason, I strongly recommend the authors post these data on a FAIR-certified public repository prior to publication.

Data and methodology:

One clarification is required: in Extended Data File 3, Food Deficit & Supply, sources cited include links to ourworldindata.org. However, this website aggregates other publicly available data, and provides it in a GUI format. Please provide the link to the original data underlying the Our World In Data GUI. Further, the link provided for the Food Deficit data does not go to a page that includes Food Deficit. It looks like maybe the authors calculated the deficit based on the difference between minimum daily requirements and country-level calories available per capita. If this is the case, it needs to be described.

Analytical approach:

The analytical approach is sound. I am not an expert in land cover classification methods, but the considerations the authors listed are thorough, and the sources of their methods are current.

Suggested improvements:

In addition to my suggestion about changing the results from impacting country- and global-level food deficits to just calories produced and population potentially fed by those calories, I have a few suggestions for improvements.

In Figure 3, it is important for the reader to know that the non-spatially optimized scenarios are generated with random assignment. This information needs to be mentioned earlier in the main text, before the first reference to the figure. Also, the methods text on lines 621-624 should be moved up to the main text so readers can fully understand Figure 3.

Extended Data Figure 5 is very interesting and has a significant value to the focus of the paper. This figure deserves a place in the main text, possibly more so than Figure 4 which I find has the problematic country- and global-level food security results.

The sentence on lines 207-209 is confusing.

Global maps make it difficult to see details at policy-relevant scales (usually country-level or finer). It would add value to provide at least continent-level maps with country borders drawn to the Extended Data Figures section.

Clarity and context:

The text is written clearly, and enough context is provided to understand the relevance of the study. I suggest less space be dedicated to the Tradeoffs and Synergies section, as these scenarios are all highly hypothetical and not possible due to lack of global cooperation. This would allow more space in the main text to include details currently in the Methods on a few key items that provide clarity, specifically: the methods used to identify abandoned cropland, methods for estimating managed reforestation, and the results shown in Extended Data Figure 5.

Reviewer #3:

Remarks to the Author:

Dear authors, I am honored to review your work. Focusing on the food security effects and carbon sequestration potential of abandoned farmland is an interesting study. The authors assess the benefits of re farming and afforestation scenarios for abandoned farmland. The article has publication potential. However, there are still some questions to ponder before publication. Specific suggestions are as follows:

(1) How do the authors distinguish between land that can be re farmed and land suitable for forest restoration? This information needs to be stated in detail.

(2) The authors consider too few factors when considering the suitability of abandoned farmland. It has been mentioned in existing studies that disaster is one of the important driving factors affecting farmland abandonment. I suggest the authors take regional disaster as a potential driving factor for re farming.

(3) The authors refer to the carbon sequestration rate of active reforestation. In this case, active reforestation means what stage is the forest in? Seedling stage or maturity stage? The carbon sequestration rates of different growth cycles are different, which cannot be compared with the current land cover conditions. Therefore, I think the study may overestimate or underestimate the carbon sequestration potential.

(4) Line 3-4, the authors have access to the most recent data, detailed in the State of Food Security and Nutrition in the World 2022.

(5) Line 35-37, this sentence is very confusing to me. In general, the impact of abandoned farmland on food security is negative. The authors say the link between abandoned farmland and food security is unclear, and some evidence is needed.

(6) When considering the multi-use scenario, do the authors consider the use scenario that "food production can both produce carbon emissions and achieve carbon sequestration"? Abandoned farmland has an intersection in food production and climate change mitigation. For example, food production can both produce carbon emissions and achieve carbon sequestration. I suggest that the authors further analyze and study the part of "Trade-offs and synergies" to increase the depth of the article.

Thank you for your efforts and look forward to your reply.

RESPONSE TO REVIEWER #1

General comment

Thank you, authors, for this study. The study aims to fill an important gap in the literature by comparing the global benefits of recultivation of abandoned cropland for food production and alternative land-based climate change mitigation through active afforestation at high spatial resolution. I found the manuscript novel and well-written and enjoyed reading it. The topic should be of broad interest. However, I also found that the present manuscript falls short of addressing this important topic in a convincing way. I have several major methodological concerns that should be solved, such as related to the quantification of active reforestation rates, lack of soil carbon consideration, and unclear time horizon. The analysis does not currently reach its untapped potential. I hope my suggestions can help the authors strengthen their study further.

Main strengths:

1. The manuscript critically advances the literature by spatially quantifying recultivation potentials for food production and achievable climate change mitigation of active reforestation at a global scale.
2. It proposes prioritization strategies for the management of abandoned croplands that may help achieve global sustainability goals. This is a much-needed contribution and provides compelling insights.
3. A wide range of models/data sources are integrated to address the research questions at hand in a new manner.
4. The paper is overall well-written.

Best regards,
Jan Sandstad Næss

Response

Dear Dr. Jan Sandstad Næss,

We are grateful for your insightful and encouraging comments, which helped a lot in strengthening our study. Following your suggestions, we have revised the manuscript substantially. Specifically, we have addressed methodological issues, edited texts to make our manuscript clear and informative, and performed a series of additional analyses, together with new supplementary figures, tables, data and documents. Our point-by-point responses are listed below. The corresponding revisions are marked in **green color** in the revised manuscript.

Best regards,
Qiming Zheng

Comment 1

Climate change mitigation. It should be clearer from the beginning that the “mitigation” scope of this paper is active reforestation as an alternative to natural regrowth. Reading the title and the abstract now, there is no way to identify that you are addressing active reforestation, and

not one of the many other land-based mitigation measures available as future trajectories for abandoned cropland. This should be clarified both in the abstract and title.

Response

Thank you for your comment. We agree with you that we need to make it clear about the active reforestation approach. We have revised the manuscript accordingly. We have also followed your **Comment 4(v)** to “drop the conversion factor altogether in the spatial analysis, and instead change the research questions towards finding priority areas for natural regrowth using Cook-Patton dataset.” We now clearly indicated the “mitigation” scope of our paper is reforestation via natural regrowth in the Abstract (Line 3-5), Main text (Line 73-76, Line 139-142, and many other places), Discussion (Line 368-370), and Methods sections (Line 615-647).

Comment 2

Carbon fluxes over time. Arguably, what is here called “baseline carbon sequestration conditions” (e.g., natural regrowth) is already an ongoing process that removes carbon dioxide from the atmosphere over time relative to the previous land use (cropland). The authors point out that the “additionality” relative to baseline conditions is the only mitigation that is tradable as carbon credit. Therefore, the difference between active reforestation and natural regrowth is quantified as mitigation. This thus raises the question, what is in fact the business as usual?

Response

Thank you for your comment. We agree with your point that definition of “additionality” would raise confusion on what is “business as usual”.

Please note that we have followed your **Comment 4(v)** to drop the analysis with active reforestation. In this case, there is no unclear issue regarding “additionality” and “business as usual”.

Your results indicate minor recultivation activity (Ext. Data Fig. 6). However, according to Crawford et al. (2022) that considered satellite observations from Landsat at 30m resolution (85% overall accuracy claimed), cropland abandonment is highly ephemeral and more than half of historical cropland abandonment in abandonment hotspots have typically been recultivated within 30 years. So, who has the more reliable data? There is clearly a disagreement there. If the findings of Crawford et al. are correct, does it still make sense to rely on continued natural regrowth as a baseline and only show the difference? Or is avoiding recultivation in fact, already an intervention requiring active policy measures?

Response

Thank you for your comment. First, we would like to highlight that the key take-home message of Crawford, et al. ¹ is that abandonment can be ephemeral. Their findings emphasize the importance of land-management interventions of recultivation or reforestation as an approach to circumventing this ephemeral nature of abandoned croplands, otherwise, as you mentioned, it will lead to the loss of climate change mitigation potential. Our study expands their findings by identifying land-use management strategies to better utilize abandoned cropland.

Second, recultivated abandoned cropland has been masked out in our analysis, therefore we only focus on the permanent abandonment cropland.

Third, the differences in the amount of recultivated abandoned cropland between our study and Crawford, et al. ¹ can be ascribed to the following reasons:

- (1) **Different spatial resolution of input data** (300-m ESA-CCI vs. 30-m Landsat). The scale effect is one of the most common reasons causing different findings for studies based on remote sensing products². Lu, et al. ³ showed that cropland maps derived from datasets with different spatial resolutions can lead to distinct spatial patterns, therefore making the subsequent findings not always comparable. In addition, the relative coarse spatial resolution of ESA-CCI product may also play a part as cropland changes (e.g., abandonment and recultivation) occurring within each 300-m ESA-CCI pixel can be under-detected.
- (2) **Different study areas**, i.e., ours (global) vs. Crawford, et al. ¹ (11 study sites, ~180km*180km each). Some of these study sites only have a small amount of cropland abandonment (e.g., Mato Grosso, Brazil, abandonment extent/cropland extent=1.32%) and only one with widespread abandonment (cross-border Belarus/Russia with little recultivation on the Russian side). Other cropland abandonment hotspots are not included in these 11 study sites, such as Southeast Asia, Central America, and Central Africa⁴⁻⁶. While they applied elaborative methods to investigate abandonment and recultivation in these 11 study sites, what happened to these 11 study sites might not be consistent with what happened at a global scale.
- (3) **Different definitions of cropland and abandoned cropland** (i.e., semantic difference). Studies have found that different definitions of “forest” would lead to different outcomes of forest change mapping^{7,8}, and consequently cause cross-study inconsistency.

Similarly, the difference in how “cropland” is defined between our study and Crawford, et al. ¹ could lead to differences in the identified cropland change patterns (i.e., abandonment and recultivation). Specifically, the cropland map used in our study follows ESA-CCI and IPCC land category schemes⁹, which include mosaic cropland/natural vegetation as a “cropland” class. In comparison, the input cropland map for Crawford, et al. ¹ is derived from Yin, et al. ¹⁰, which does not specify how to define and deal with such mosaic land cover.

As for the definition of cropland abandonment, our study excludes built-up areas and human settlements when identifying abandoned cropland⁴, while the abandoned cropland maps used in Crawford, et al. ¹ seem not to exclude built-up areas and human settlements. By overlying the abandoned cropland maps used in Crawford, et al. ¹ with Google Earth images, we observed that many roads and human settlements were classified as abandoned cropland (**Fig. R1**).

[redacted]

Fig. R1 | Two examples of the original abandoned cropland maps used in Crawford, et al. ¹.
Data source: <https://zenodo.org/record/5348287#.ZBiOXXZByUn>.

Yet, our discussions neither mean to undermine the significance of Crawford, et al. ¹ nor to say whose results are correct because the above-mentioned differences between two studies hamper a quantitative and fair comparison. Instead, our discussions aim to explain the possible reasons for the differences between two studies. We acknowledge this valuable research in our study because it is pointing to the potential threat from recultivation of abandoned cropland and the consequent impacts on its climate change mitigation potential.

In addition, we compared the recultivation rates of 11 study sites of Crawford, et al. ¹ (Fig. R2b) with the historically recultivated abandoned cropland identified in our study (Fig. R2a). We found that the spatial patterns of recultivation identified in both studies are generally consistent, albeit with different recultivation intensity.

In the revised manuscript, we have (1) described the linkage between between our study and Crawford, et al. ¹, and (2) added a detailed discussion to let readers know the possible reasons leading to the differences between two studies.

Please see Line 60-64, Line 571-573, and Supplementary Text 8.

[redacted]

Fig. R2 | Comparison between the recultivated abandoned cropland identified in our studies (a) and Crawford, et al. ¹ (b). Fig. R1b and the bar-plot are adapted from Fig. 1 and Fig. S4 of Crawford, et al. ¹. Data source: <https://zenodo.org/record/5348287#.ZBiOXXZByUn>.

I suggest a change in thinking towards including negative emissions over time of continued natural regrowth in some of your results. It would improve the overall comprehensibility to show average yearly carbon fluxes over a given period (for example, 30 years based on Cook-Patton data limitations) simultaneously for both natural regrowth, active reforestation, and recultivation. The additionality of carbon dioxide removal contribution from active reforestation will be visible. See for example, Field et al. Figs 3 and 4 for example comparison of continued regrowth/bioenergy/BECCS.

Response

Thank you for your suggestion. We have added a figure to show the breakdown of climate change mitigation potential over time from recultivation (including due to emissions of land clearing and foregone potential from natural growth) and forest restoration via natural regrowth (Fig. R3). We do not include active restoration here because we changed our analysis to natural regrowth as per your **Comment 4(v)**.

Please see Supplementary Text 2 for methodological details.

Fig. R3 | Breakdowns of the climate change mitigation potential, including carbon sequestration obtained from reforestation via natural regrowth, emissions from clearing the

historically accumulated biomass carbon for reclamation (without energy recovery), and the lost mitigation potential of foregone natural regrowth if the abandoned cropland is reclaimed.

Comment 3

Climate impacts of reclamation. It is inconsistent not to quantify the climate impacts of reclamation and on-farm activities when designing prioritization strategies between climate change mitigation and reclamation for food production.

Emissions related to land use change (i.e., the clearance of historically accumulated aboveground carbon) are quantifiable by the datasets you already have at hand.

Furthermore, life cycle emissions related to on-farm activities such as usage of machinery, fertilizer, pesticides etc. and supply chain emissions should be accounted for.

The lost mitigation of foregone natural regrowth should also be shown.

Response

Thank you for your insightful comments.

We have quantified the impact of future climate change using the projected attainable yields under RCP4.5 and RCP8.5 scenarios and during 2010-2040 and 2040-2070. The future attainable yield of each crop is the ensemble mean of five climate model outputs produced by GAEZ v4 (Table R1). We have also shown how future climate change affects spatial prioritization of reforestation and reclamation (the bottom right of Fig. R4).

Please also see Line 329-332 and Supplementary Text 3.

Table R1 | Impact of future climate change on crop yields and achievable food production potential on current abandoned cropland (original estimate based on rain-fed actual yield of 2010: 363 Pcal yr⁻¹).

Climate forcing	Water supply	2010-2040	2040-2070	Diff.
RCP4.5	Rain-fed	506 Pcal yr ⁻¹	522 Pcal yr ⁻¹	+3%
	Irrigated	728 Pcal yr ⁻¹	814 Pcal yr ⁻¹	+12%
RCP8.5	Rain-fed	508 Pcal yr ⁻¹	523 Pcal yr ⁻¹	+3%
	Irrigated	749 Pcal yr ⁻¹	823 Pcal yr ⁻¹	+10%

Fig. R4 | Recultivation priorities when different yield improvement approaches are applied.

Following your suggestion, we have taken the emission related to land clearing into consideration. We calculated the historically accumulated aboveground biomass carbon using Cook-Patton, et al.¹¹ and the biomass carbon stock map from Spawn, et al.¹², and estimated the carbon emission if all the existing aboveground carbon is released to the atmosphere (i.e., without energy recovery).

All the related results have also been updated, including Figs. 1-4, Table 1, and content in Supplementary Materials. Consequently, we found that emissions from land clearing would offset the net climate change mitigation potential from 0 MtCO₂ yr⁻¹ (“climate change mitigation only” scenario) to -156 MtCO₂ yr⁻¹ (“food production only” scenario).

Please see the breakdowns of the climate change mitigation potential in the figure below, including the lost mitigation of foregone natural regrowth. More detailed figures and descriptions are presented in Table 1, Line 655-669 and Supplementary Text 2.

Fig. R3 | Breakdowns of the climate change mitigation potential, including carbon

sequestration obtained from reforestation via natural regrowth, emissions from clearing the historically accumulated biomass carbon for recultivation (without energy recovery), and the lost mitigation potential of foregone natural regrowth if the abandoned cropland is recultivated. Also see Supplementary Fig. 7.

We agree that it is important to incorporate life cycle emissions into our analysis. We encountered two hurdles. First, the life cycle emissions are not available for all 15 crops considered in our study, while the life cycle emissions are expected to vary across crops due to different fertilizer demands and management strategies (e.g., rice vs. wheat)¹³. Second, the available life cycle emission data are mostly global estimates and the life cycle emissions should be spatially varied, e.g., due to different transport distances and fertilizer use efficiency, thus making it unsuitable to be used for spatial prioritization analysis. We have highlighted these limitations. Please see Line 349-356 and Line 666-669.

Comment 4

Active reforestation carbon accumulation (key point, hence long comment!). The Cook-Patton machine-learning dataset is a robust machine-learning dataset providing natural regrowth rates at high resolution and likely the best one available. This study makes use of a global conversion factor to spatially quantify active reforestation by multiplying grid-specific natural regrowth rates for all pixels with the conversion factor. This conversion factor is based on relatively few studies (n=27, Supplementary Data 2), and the spatial locations of observations are heavily centered in a few countries (mainly in the tropics). Thus, applying this global factor to a gridded abandoned cropland map with a different spatial distribution of available land means you are extrapolating the average global factor into locations that are not covered by your gathered data. Does this even make sense? This degrades the initial robustness coming from the Cook-Patton dataset, that is based on a much larger set of observations and more sophisticated modeling techniques (although data gaps are also present there). The conversion factor variation is also large, even within countries. Furthermore, it is not shown if factor variation arises due to chosen tree species, different on-site management, or different local climatic conditions. I don't find it sufficient to only explore uncertainty through the standard deviation of variation arising from this limited information from 27 studies, mainly due to the geospatial variation of abandoned cropland intensity. Also, the major uncertainty shown undermines the significance of the study. I'd suggest the authors to consider the following actions:

- i) Compare the mapped active reforestation carbon accumulation rates with output from other models. Preferably, this should include a spatial component and cover either the geographical regions, climate zones, biomes, or plant functional types that are dominant in the areas with most intense cropland abandonment (e.g., Eastern Europe, Brazil, Central Africa, Indonesia, Central America). Models that could be compared against include LPJmL, CLM, or G4M ++ (and yes, these models also have their own uncertainties, prediction errors, and limitations). On the top of my head, I can suggest checking Doelman et al. (2020) that has some regional/country insights and Braahekke et al. (2019). There should be much more literature out there.
- ii) Gather more data to support their modelling effort. Show how the conversion factor correlates with climatic conditions (e.g., temperature/precipitation) and active/passive management, and/or how it varies across broadleaf/evergreen forests, geographical regions,

biomes, or climatic zones, and assessed time horizon. Use these insights to improve their model.

iii) Gather observational data on aboveground carbon accumulation from active reforestation projects for the sole purpose of validation. Compare predicted carbon accumulation rates from the active reforestation map with observations at a locational basis and produce some statistical insights.

iv) Instead of relying on the combination of a machine-learning dataset and a simplified conversion factor, run a process-based land surface model to quantify active reforestation carbon accumulation.

v) Drop the conversion factor altogether in the spatial analysis, and instead rely directly on the Cook-Patton dataset. This might mean changing the research question towards finding priority areas for continued natural regrowth instead of for active reforestation. This still provides a novel contribution. The global average conversion factor can still be used to show an aggregated global mitigation estimate of intervening through active reforestation across scenarios, but then without presenting spatial results and whilst clearly explaining the underlying limitations.

Response

Thank you for your insightful comments! We agree with your concerns about our estimation of the carbon sequestration rate of active reforestation. According to your **Comment 4(v)**, we have changed our analysis into directly relying on Cook-Patton's carbon sequestration rate data. Consequently, our studies turned to comparing the trade-offs between recultivation and forest restoration via natural regrowth.

We have updated all the related content, including but not limited to the achievable climate change mitigation potential, trade-off simulation and analysis, and spatial prioritization. Our updated results showed that reforesting abandoned cropland via natural regrowth can still contribute a considerable amount of climate change mitigation potential, although it is a bit lower than active reforestation. If we reforest all suitable abandoned cropland via natural regrowth, it could sequester carbon for 1,080 MtCO₂ per year. This climate change mitigation potential could help to offset 3-7% of the global emission reduction needed for achieving the 2°C climate goal (SSPs-26, 2020-2050)¹⁴.

Please see Line 139-157 (Main results) and Line 662-667 (Methods).

Besides, we followed your suggestion to provide an aggregated global estimate of the climate change mitigation potential of active reforestation using our conversion factor instead of a spatially-explicit map. Our estimation indicates an increase of 53% in the achievable climate change mitigation potential if we actively reforest abandoned cropland outside protected areas (defined by IUCN and UNEP¹⁵, also see Supplementary Fig. 11). We have also discussed the underlying limitations of such estimation approach.

Please see Line 336-348 and and Supplementary Text 3.

Comment 5

Soil carbon change. There is no consideration or even discussion of soil carbon implications of the different land uses you propose. An attempt to quantify this would be beneficial. Some discussion is a necessity. Cook-Patton et al. shows some belowground carbon accumulation rates, both as global averages and biome-specifics (although, large variation even within biomes). Bell et al. is relevant. Please review some literature.

Bell, S. M., Barriocanal, C., Terrer, C., & Rosell-Melé, A. (2020). *Environmental Science & Policy*, 108, 104-111. <https://doi.org/10.1016/j.envsci.2020.03.018>

Response

Thank you for providing this important literature, we have included it in the revised manuscript. We chose not to include soil carbon because the soil carbon accumulation rates after natural regrowth were found negligible or negative, even contrasting across studies^{11,16,17}. Unlike changes in aboveground and belowground biomass carbon, soil carbon change during natural forest regrowth is understudied and short of measurements. We have added an explanation of why soil carbon was not considered (Please see Line 642-645).

Comment 6

Aboveground carbon stocks on abandoned cropland were not quantified. Having an idea of the historical carbon accumulation on the abandoned cropland is important for a couple of reasons. i) reclamation for food production means there is a need to clear the secondary vegetation. If this biomass is combusted this creates a non-neglectable initial emission pulse that contributes to radiative forcing and that should be quantified and accounted for. ii) Aboveground carbon may also need to be cleared before tree planting depending on the local conditions.

This has already been done at least twice. First by Crawford et al. (2022) in abandonment hotspots by integrating Landsat and Cook-Patton, then by Næss et al. (2023) for the Nordics by integrating ESA CCI-LC and Cook-Patton. The authors clearly have the data at hand to do the same, and I see no scientific reasons not to consider this as it has been shown to be important. Especially, the fate of this biomass is also important, whether it is burned at site without energy recovery, combusted with energy recovery that may replace fossil fuels (e.g., wood stoves, or fed to modern bioenergy supply chains to produce biofuel, bioelectricity, or hydrogen), or even used as feedstocks to BECCS. As a minimum, I'd suggest considering that aboveground carbon is combusted at site without energy recovery. The time horizon also matters for the relative importance of clearing aboveground carbon. Longer time horizons mean it gets less important.

Response

Thank you for your suggestions. For reclamation, we modified our analysis by taking the emissions from clearing the historically accumulated aboveground biomass carbon on abandoned cropland into consideration. We calculated the historically accumulated aboveground biomass carbon using Cook-Patton, et al. ¹¹ and the biomass carbon stock map from Spawn, et al. ¹², and estimated the carbon emission if all the existing aboveground carbon is released to the atmosphere (i.e., without energy recovery).

Please see the breakdowns of the climate change mitigation potential in figure below, including the lost mitigation of foregone natural regrowth. More detailed figures and descriptions are

presented in Supplementary Text 2.

Fig. R3 | Breakdowns of the climate change mitigation potential, including carbon sequestration obtained from reforestation via natural regrowth, emissions from clearing the historically accumulated biomass carbon for recultivation (without energy recovery), and the lost mitigation potential of foregone natural regrowth if the abandoned cropland is recultivated. Also see Supplementary Fig. 7.

We also provided a discussion on the potential benefits of energy recovery to offset emissions from land clearing (Please see Line 478-482). Again, since we changed to forest restoration via natural regrowth (refer to **Response to Comment 4v**), land clearing is not necessary¹⁸. We have clarified the time horizon of our study (i.e., from 2020 to 2050). Please see Line 631-634.

Comment 7

The considered time horizon is unclear. Results are frequently presented on a per year basis without clarification of considered period. The used datasets limit how far into the future you can claim to assess. The Cook-Patton dataset considers 30 years. GAEZ typically predicts 30-year average food crop yields depending on background climate. Actual historical yields from GAEZ may be based on shorter time horizons (e.g., single years). Hanssen et al. (2020) shows how important chosen time horizon can be when assessing climate change mitigation potentials of negative emission technologies. The importance of time horizon when assessing carbon dioxide removal is also highlighted by Terlouw et al. (2021).

-Hanssen, S.V., Daioglou, V., Steinmann, Z.J.N. et al. The climate change mitigation potential of bioenergy with carbon capture and storage. *Nat. Clim. Chang.* 10, 1023–1029 (2020). <https://doi.org/10.1038/s41558-020-0885-y>

Terlouw, T., Bauer, C., Rosa, L., & Mazzotti, M. (2021). Life cycle assessment of carbon dioxide removal technologies: a critical review. *Energy & Environmental Science*, 14(4), 1701-1721. <https://doi.org/10.1039/D0EE03757E>

Response

Our time horizon is the next 30 years since 2020 (i.e., 2020-2050). In our study, we used Cook-Patton’s carbon sequestration rate of forest nature regrowth, which is a 30-year estimate¹¹. We used the rain-fed actual yield of 2010 to estimate the food production potential during 2020-2050. While we admitted that it would underestimate the food production potential, the following two concerns made us use actual yields instead of future attainable yields in our main

analysis:

- (1) **It is even unlikely to achieve attainable yields on abandoned croplands.** Attainable yield is the yield achieved through skillful use of the best available technology and management (as defined by FAO¹⁹). However, such optimal technology and management are unlikely to achieve because many cropland abandonment spots are distant from cities, with a shortage of labor and equipment and less favorable agro-climate conditions^{20,21}.
- (2) **Only high-input yield estimates are available for future attainable yields (e.g., 2010-2040 and 2040-2070).** High-input yields are based on advanced management assumptions. The cropland is assumed to be fully mechanized, with reduced but well-trained labor and optimum applications of nutrients and chemicals for pest, disease and weed control. Due to the same reason in the above-mentioned Point 1, we consider it would be more realistic to use the actual yield for food production estimation.

To address the above-mentioned issues, we have (1) clarified the time horizon of our studies (i.e., 2020-2050) (see Line 478-482 and Line 604-605); (2) indicated which yield data we used and explained the rationale (see Line 597-601); (3) added a detailed discussion on the fact that our estimated food production potential was very conservative (Line 304-334); and (4) provided the estimated outcomes using future attainable yields (2010-2040 and 2040-2070, ensembled from five climate models under RCP4.5 and RCP8.5 scenarios) (Table R1 and Fig. R5). By running the comparison of these outcomes, hopefully we can give readers a better sense of the possible potential food production increases due to agricultural intensification, management improvements and climate change.

Please also see Fig. 4, Line 732-750, and Supplementary Text 3.

Table R1 | Impact of future climate change on crop yields and achievable food production potential on current abandoned cropland (original estimate based on rain-fed actual yield of 2010: 363 Pcal yr⁻¹).

Climate forcing	Water supply	2010-2040	2040-2070	Diff.
RCP4.5	Rain-fed	506 Pcal yr ⁻¹	522 Pcal yr ⁻¹	+3%
	Irrigated	728 Pcal yr ⁻¹	814 Pcal yr ⁻¹	+12%
RCP8.5	Rain-fed	508 Pcal yr ⁻¹	523 Pcal yr ⁻¹	+3%
	Irrigated	749 Pcal yr ⁻¹	823 Pcal yr ⁻¹	+10%

Fig. R5 | Possible approaches to improve achievable food production potential and climate change mitigation potential or to free up land for reforestation (or recultivation) while still achieving the original food production outcomes (or climate change mitigation outcomes). The error bars indicate the mean \pm standard deviation of the additional potential obtained from the freed-up abandoned cropland. Results are based on “maximizing food production” scenario and “maximizing climate change mitigation” scenario, with all suitable abandoned cropland used. Attainable yields are the 30-year mean of 2010-2040 under RCP4.5 condition (see Supplementary Text 3). PA: protected areas defined by UNEP and IUCN.

Comment 8

GAEZ crop yields. The methodological description made me confused (lines 478-506). It is written that present day actual rain-fed yields at low agricultural input as 30-year means (1980-2010) are considered. In GAEZ v4, there are three types of yield outputs, i.e., agro-climatic yields, attainable yields, and actual yields. Actual yields are derived through downscaling annual national average yields over 3-year periods (for example, for 2010 based on the years 2009-2011) from agricultural statistics (FAOSTAT). Only the GAEZ modelled agro-climatic yields and attainable yields are representative of 30-year averages, and they thus rely on different background climates (e.g., historical climatic conditions from CRUTS-32, or for future RCPs from Earth System Models). The actual yields are also not modelled based on agricultural input (as claimed in the manuscript), this is only the case for agro-climatic and attainable yields. Please clarify better what was used and ensure that the methodological description aligns with the GAEZ model documentation. If you considered attainable yields, then make sure to describe and cite the correct background climate dataset/model.

Response

Sorry for the confusion. We have (1) corrected the description of actual yield (Line 597-601); and (2) clarified which yield data were used in our study, as well as the associated time horizon (Line 604-605), input-level setting (Line 315-318), and background climate conditions (Line 737-743).

Please also see Table R2 below and Supplementary Text 3.

Table R2 | Yield data used in this study and the associated settings.

Yield type	Period	Water supply	Input level	Climate data source	Climate Forcing
Actual yield (Main analysis)	2010	Rainfed Irrigated	NA	CRUTS-32	Historical
Attainable yield (Supplementary analysis)	2010- 2040	Rainfed Irrigated	High*	Ensemble (Ensemble mean of all models' outputs)	RCP 4.5 RCP 8.5
Attainable yield (Supplementary analysis)	2040- 2070	Rainfed Irrigated	High*	Ensemble (Ensemble mean of all models' outputs)	RCP 4.5 RCP 8.5

*: Only yields with high agricultural input (high level inputs) are available in the GAEZ database. High level inputs refer to the condition that the farming system is mainly market oriented with advanced agricultural management. Production is based on improved or high yielding varieties, is fully mechanized where possible with low labor intensity and uses optimum applications of nutrients and chemical pest, disease and weed control²².

Comment 9

It is claimed that GAEZ “low agricultural input” is considered in the main analyses, but it is never explained what this means, and it needs to be clarified. To quote the GAEZ model documentation:

“Low-level inputs/traditional management

Under the low input, traditional management assumption, the farming system is largely subsistence based and not necessarily market oriented. Production is based on the use of traditional cultivars (if improved cultivars are used, they are treated in the same way as local cultivars), labor intensive techniques, and no application of nutrients, no use of chemicals for pest and disease control and minimum conservation measures.”

Considering yield predictions based on no mechanization, no fertilizer use, no pesticides, etc., will likely produce an underestimation of recultivation potentials in developed countries. It is acknowledged that estimates are conservative (e.g., underestimation). GAEZ has the capability of providing yield predictions for more modern agricultural management conditions. Why not use the model’s capabilities to produce estimates of attainable yield reflecting actual

management variation? Considering a spatially differentiated management factor accounting for varying yield gaps in combination with high/low input yields from GAEZ can provide insights. Yield gaps are available from the GAEZ database.

Response

Thank you for your suggestions. We have added definitions of input levels based on GAEZ v4 document²². Please see Line 315-318.

We have added an explanation regarding why we used actual yields for our main analysis. Please refer to our **Response to Comment 7**. Besides, we have followed your suggestion to estimate the outcomes if we improve the crop yields to attainable yields for areas with yield gaps identified from the GAEZ database (Supplementary Figs. 8&9). We found that it would increase the achievable food production potential by 40% (attainable yield + high input) and 101% (attainable yield + high input + irrigation) compared with our original estimates. Alternatively, if we maintain the food production at our original estimate, yield improvements would free more land for reforestation and generate additional climate change mitigation potential by 43% (attainable yield + high input) and 76% (attainable yield + high input + irrigation) (see Fig. R5 below).

Please also see Fig. 4, Line 304-334, and Supplementary Text 3.

Fig. R5 | Possible approaches to improve achievable food production potential and climate change mitigation potential or to free up land for reforestation (or reclamation) while still

achieving the original food production outcomes (or climate change mitigation outcomes). The error bars indicate the mean \pm standard deviation of the additional potential obtained from the freed-up abandoned cropland. Results are based on “maximizing food production” scenario and “maximizing climate change mitigation” scenario, with all suitable abandoned cropland used. Attainable yields are the 30-year mean of 2010-2040 under RCP4.5 condition (see Supplementary Text 3). PA: protected areas defined by UNEP and IUCN

Comment 9

Extended Data fig 5 provides valuable insights into how both improved management and irrigation might increase potentials and help free land for reforestation. I found this figure very helpful. However, I was left wondering how this affects the spatial prioritization. Maybe you can include a SI figure showing this. Also, please describe in the methods what high inputs means in GAEZ (see GAEZ documentation cited above).

Response

Thank you for your suggestions. We have (1) added the spatial prioritization maps with improved management conditions, water supply, and yield types (Fig. R4 below); and (2) added a description of high-level inputs based on GAEZ v4 document (Line 315-318).

Please also see Supplementary Text 3.

Fig. R4 | Recultivation priorities when different yield improvement approaches are applied.

Comment 10

Missing insights into crop distribution. 15 food crops are considered based on historical allocation patterns. However, there is no transparency into their spatial deployment, the extent of land allocated to different crops, or what crops provide the dominant contribution to production potentials. Which crops are the important ones? Presenting a map of dominant crop

allocation and quantifying these things would help the reader's understanding. What about the effect of optimizing the crop distribution for maximized calorie production? Would this yield a substantial gain in potential?

Response

Thank you for your suggestions. The impact of optimizing crop distribution on maximizing calorie production can lead to a substantial increase in the achievable food production potential on abandoned cropland. For example, under rain-fed and present-day (2010) actual yield conditions, optimizing crop distribution would increase food production potential by 74% compared with the crop allocation based on the current harvest area (also see Line 332-334).

In the revised manuscript, we have: (1) added crop allocation maps of each food crop and an integrated allocation map indicating the distribution of the dominant crop (Fig. R6); (2) added simulated outcomes with an optimized crop allocation for maximizing calorie food production (Supplementary Table 2); and (3) added the crop allocation maps with optimized crop allocation (Fig. R6).

Please also see Supplementary Text 3.

Fig. R6 | The spatial allocation map of 15 selected crops (%) and the corresponding dominant

crop types of each 5arcmin pixel (under rain-fed and present-day actual yield conditions). The crop allocation is based on current harvest area of each crop. The crop allocation that maximizes calorie food productivity is also provided.

Comment 11

Lack of validation of crop yields. Have GAEZ predicted attainable food crop yields been validated against observations? If yes as I would expect, it seems appropriate to provide some indication of performance with associated references. Otherwise, providing some validation with comparison of predictions and observations might be necessary.

Response

Crop yields produced by top-down approaches, including the GAEZ model, are difficult to validate directly by observational data because the results of these approaches are necessarily aggregated to the grid level, without differentiating amongst soil types and cropping systems that may exist within the grid²³.

However, the GAEZ model and its estimated crop yields have been validated by (1) comparisons to other model outputs^{24,25}, (2) comparisons to statistical records²⁶, and (3) comparisons against crop yields map estimated by bottom-up approaches (i.e., models based on local observations), such as GYGA data^{23,27}. These validation efforts have shown an acceptable and conservative performance of crop yields estimated by GAEZ. GAEZ has been one of the most widely-used data for global scale crop yields estimate²⁸⁻³⁰.

Following your suggestions, we have added a brief introduction in the Method section about how crop yields estimated by GAEZ have been validated. Please see Line 587-593.

Comment 12

Abandoned cropland validation. While ESA CCI-LC has been used before to map abandoned cropland (e.g., Næss et al. (2021) and Leirpoll et al. (2021)), this study aims to provide for the first time an accuracy assessment of an ESA CCI-LC based abandoned cropland map (80% overall accuracy claimed) instead of relying only on cropland class accuracies and cross-study comparisons. This is an important contribution. However, it is currently not transparent enough to evaluate it properly. The claim should be supported by shown data.

First, I was left wondering why the underlying confusion matrices were not provided in a supplement as for example done in the SI of Lesiv et al. (2018)? It is recommended by Olofsson et al. (2013) to include them.

Response

Thank you for your suggestion. We now provide the confusion matrices and accuracy assessment scores, including overall, producer's, user's accuracies, and F1 score (Table R3). We also reported the confidence interval of the total abandoned cropland extent estimated in our study with the accuracy assessment method proposed in Olofsson, et al. ³¹.

Please refer to Supplementary Text 6 for details.

Table R3 | Confusion matrix at the global level.

	Reference	
	Abandoned cropland	Non-abandoned cropland
Abandoned cropland	773	55
Non-abandoned cropland	232	596
Accuracy assessment indices		
Overall accuracy	0.828	
F1 score	0.843	
Producer's accuracy of abandoned cropland	0.770	
User's accuracy of abandoned cropland	0.934	
Abandoned cropland extent error (95% CI) ³¹	±35 Mha	

Second, your sample size also seems relatively small (n=500), whilst for example Lesiv et al (2018) used about 6000-pixel samples in their abandoned cropland validation. I don't see any justification for why the current sample size is sufficient in this case or discussions regarding sample size. Also, please provide the coordinates and classification of your sample in the supplementary data as done by Lesiv et al.

Response

Following your suggestions, we have provided the locations and classification of our validation samples (Fig. R7). We also increased the number of validation samples to n=1656. We first followed Eq. 13 of Olofsson, et al. ³² to determine the initial sample size for abandoned cropland class (57 samples) and non-abandoned cropland class (828 samples). Then, to avoid problems with small sample sizes for rare thematic classes (abandoned cropland), we used disproportionate stratified sampling³³ by selecting 828 samples for both abandoned cropland class and non-abandoned cropland class, thus in total 1656 validation samples.

Please refer to Line 525-531, Supplementary Fig. 13 and Supplementary Text 6 for details.

We also qualitatively compared the spatial patterns of abandoned cropland identified in our study with previous studies (n=42; Supplementary Data 5).

Fig. R7 | Spatial distribution of validation samples.

Third, it would be beneficial to bring up estimated producer and user accuracies of the abandoned cropland class in the main text for enhanced understanding. Could be compared with previous studies such as Lesiv et al. Overall accuracy seems not enough.

Response

As per your suggestion, we have added the producer’s, accuracy, user’s accuracies, and F1 score in the main text, together with the overall accuracy. Please see Line 93-96.

Fourth, did you find any geospatial (i.e., for different regions or climate zones) or temporal variation in abandoned cropland accuracy? Such spatiotemporal insights of map accuracy would be very helpful when assessing the reliability of the abandoned cropland map at a continental, regional, or local level.

Response

We have provided the accuracy assessment results for each continent. We found the mapping overall accuracies of Africa (0.76) and Europe (0.79) are slightly lower than other continents (Table R3).

Please also see Supplementary Table 6 in Supplementary Text 6.

Table R3 | Confusion matrix for each continent.

Continents	Overall accuracy	Producer's accuracy	User's accuracy	F1 score
Africa	0.76	0.86	0.71	0.77
Asia	0.84	0.96	0.79	0.87
Australia	0.8	NA*	0.63	NA*
Europe	0.79	0.92	0.75	0.83
North America	0.85	0.95	0.79	0.86

South America	0.88	0.84	0.80	0.82
---------------	------	------	------	------

*: Not available due to a limited number of validation samples.

Comment 13

Currently, there is no context provided as numbers regarding the future land use/cover changes associated with scenarios consistent with the Paris Agreement. It may be useful to present the afforestation extent and carbon accumulation seen in 2C scenarios towards 2050 from the Shared Socio-Economic Pathways (e.g., from Integrated Assessment Models, SSPx-RCP2.6/1.9) in the introduction. Comparing your results with such future mitigation pathways would also help make it easier to understand the potential global contribution of abandoned croplands to achieve mitigation targets.

Response

Following your suggestion, we have provided the projected forest extent and GHG emission reduction demand in 2050 under SSPs-RCP2.6 scenarios (from IIASA SSP database) and compared them with our findings to better present the global contribution of abandoned cropland.

Please see Line 152-154.

Comment 14

Recultivation for food production might help avoid cropland expansion into natural areas and prevent associated aboveground carbon losses or biodiversity impacts. It would be useful also here to link the land availability found here to SSP projections. Comparing food production potentials to increased food demand projected by IAMs in addition to the food deficits may also help.

Response

Thank you for your suggestion. Following your suggestion, we have added a comparison between future cropland expansion (SSPs-26 scenarios) and our identified abandoned cropland suitable for recultivation. However, we are not able to compare the food production potential from abandoned cropland with the food demand generated by SSP projections. This is because the IIASA's SSP database only offers future food demand projection in dry matter weight, while ours are in calorie production. Thus, we only provide a comparison between the future cropland expansion (land demand) and suitable abandoned cropland for recultivation (supply).

Please see Line 372-376.

Comment 15

Alternative land-based mitigation measures (discussion, lines 292-297). The presented argument against recultivating abandoned cropland for bioenergy deployment seems a bit of a cherry pick. Yes, the food vs fuel debate/land use competition is important, but IAMs show that bioenergy expansion is necessary to reach ambitious climate targets. The associated land use changes for bioenergy crops typically matches or even surpasses those for afforestation (see for

example, IPCC AR6, IPCC Special Report on Climate Change and Land, or any IAM paper addressing future land use). Xu et al. (2022) shows that delaying bioenergy crop deployment may in fact threaten food security as global warming causes feedback loops with associated decreasing crop yields. Arguably, reforestation and bioenergy/BECCS recultivation should be considered together, and the best use of land will likely vary based on local conditions.

Xu, S., Wang, R., Gasser, T. et al. Delayed use of bioenergy crops might threaten climate and food security. *Nature* 609, 299–306 (2022). <https://doi.org/10.1038/s41586-022-05055-8>

Response

This is really a good point! Thank you for your constructive comments. We have revised our discussion on other land-based mitigation measures by emphasizing the necessity of bioenergy expansion to reach ambitious climate targets. We have also provided the projected increase in bioenergy crop demand to meet 2°C goal with SSP2-RCP2.6 scenarios.

Please see Line 378-385.

Comment 16

A key output from this study is global priority maps. The authors claims assertively that spatially prioritized land management strategies are a necessity to make optimal use of abandoned cropland. There is an unacknowledged ongoing debate on the usefulness of global priority maps. For example, Wyborn & Evans (2021) argues that global priority maps constitute a problematic global knowledge that removes local context. Or as argued by Meyer & Pebesma (2022), global maps create a strong satisfactory feeling that can be too easily abused. This is in sharp contrast to the argument presented here. There is already some good discussion provided, but I believe that bringing your arguments into the context of this debate may help increase the impact of the study.

Wyborn, C., Evans, M.C. Conservation needs to break free from global priority mapping. *Nat Ecol Evol* 5, 1322–1324 (2021). <https://doi.org/10.1038/s41559-021-01540-x>

Meyer, H., Pebesma, E. Machine learning-based global maps of ecological variables and the challenge of assessing them. *Nat Commun* 13, 2208 (2022). <https://doi.org/10.1038/s41467-022-29838-9>

Response

Thank you for your insightful suggestions. We appreciate you bringing these papers to our attention! We agree that when implementing global-scale research, regional to local perspectives should be considered to ensure that top-down approaches do not harm or conflict with local communities. We have revised our discussion on the limitation of our global-scale findings accordingly. We begin with setting the context of a rising debate on the global map, i.e., Wyborn and Evans³⁴ (global map is problematic and will crowd out local-scale studies) and Chaplin-Kramer, et al.³⁵ (global-scale studies are necessary and as important as local studies; a correspondence paper to Wyborn and Evans³⁴). Then, we discuss important local-scale issues that should be considered to ensure the best use of the potential of abandoned cropland.

Please see Line 430-440.

Comment 17

Biophysical effects of land use changes also vary based on forest types/climate zones. I suggest to highlight how this may affect the regional climate in different abandonment hotspots in the discussion section. Cook-Patton can point you to some key papers.

Response

Thank you for your suggestion. We have added a discussion on how the regional climate (i.e., water yield, the difference between precipitation and evapotranspiration) will be affected by reforesting abandoned cropland and how the impact varies across regions.

Although many studies have documented increases in water yield from reforestation, other studies have also shown that reforestation projects would reduce water yield^{36,37}. Based on recent studies on this issue, we discussed that reforesting abandoned cropland, especially in abandonment hotspots, may lead to negative water-yield in water-stressed or semi-arid regions, such as Northern China, Middle America, East Africa, and Eastern Europe³⁸. We also underscored the necessity of avoiding causing albedo effects, particularly for reforestation practices in boreal areas, which might cause net biophysical warming³⁹.

Please see Line 459-465.

Other comments:

Comment 18

Introduction: Some more context from the SSPs on projected land use change would be helpful (show numbers).

Response

Thank you for your suggestion. We have added the following context about land-use change from 2020 to 2050 projected by SSP scenarios: (1) projected cropland expansion; (2) projected net forest expansion to meet the 2-degree goal (SSPs-RCP2.6 scenarios).

Please Line 21-24 and Line 34-37.

Comment 19

Line 76-78: Leirpoll et al. (2021) also used this dataset. Discrepancy should be mainly due to the longer period, as the cropland abandonment rate is very similar.

Response

We have revised the description. Please see Line 105-107.

Comment 20

Figure 1. The color scales are a bit hard to read without intermediate x-ticks (same with fig 2). Figure 1b seems like a binary yes/no? If that is correct, please clarify in caption.

Response

Yes, Fig. 1b presents binary suitability maps. We have modified the x-ticks for both Fig. 1 and Fig. 2.

Comment 21

Lines 299-315: This is very useful. I'd suggest also pointing out that sustainable water resources for irrigation are limited due to alternative human water usage and the need to protect environmental water flows to sustain freshwater ecosystems, but also that especially the abandonment hotspot in Eastern Europe overlaps with sustainable irrigation expansion opportunities (see Rosa et al. (2020)).

Response

Following your suggestion, we have emphasized the potential adverse impact of irrigating abandoned cropland on sustainable water resources.

Please see Line 308-312.

Comment 22

Lines 399-417: ESA CCI-LC already has a 3-year consistency rule, such that a detected land use change needs to be consistent for three consecutive years before appearing in the dataset (for example., a pixel transitioning from cropland to a non-cropland class first in 1993 needs three consecutive years detected as non-cropland before appearing in the dataset with non-cropland classification in 1995). Not sure if this was considered.

Response

Thank you for reminding us of this issue. Our five-year moving window is slightly more stringent than ESA CCI-LC's 3-year consistency rule (as described in Section 3.1.2 of ESA CCI user's document). ESA CCI-LC compares the land cover at year T with T+1 and T+2, while we compare that land cover at year T with T-2, T-1, T+1, and T+2. Similar operations (i.e., comparing the land cover type before and after the target year) have been widely used in land cover change detection studies^{1,40,41}.

We compared the total abandoned cropland area identified by applying the five-year moving window and without applying the five-year moving window (i.e., 3-year consistency rule of ESA-CCI LC only). A minor difference is observed, i.e., 101 Mha (with a five-year moving window) vs. 103 Mha (without). Such difference is mainly caused by a different land cover change detection performance in 1992-1997 and 2015-2020. As mentioned in Section 3.4 Limitation of the ESA CCI-LC user's document, "Change during the AVHRR 1992 - 1999 period: The performance of the change detection is highly dependent on the input data quality and availability. The general lower quality of AVHRR surface reflectance and georeferencing implies a less reliable change detection. In addition, the lack of AVHRR data in year 1994 reduces the change detection reliability for this particular year".

We have clarified this issue in the Methodology section. Please see Line 503-509.

Comment 23

Lines 464-466: You arrive at very different conclusions regarding the historical importance of recultivation than Crawford et al. (2022). Are there any known arguments that may cause this discrepancy (maybe resolution, CCI's 3-year consistency rule (speculating))? Cross-study comparisons and some discussion seems appropriate.

Response

Thank you for your suggestion. Please refer to our response in the **Response to Comment 2**.

In the revised manuscript, we acknowledged the difference between two studies in the main text and provided a detailed discussion and comparison two studies.

Please see Line 60-64, Line 571-573, and Supplementary Text 8.

Comment 24

Line 539: There is not any gridded belowground carbon accumulation data provided by Cook-Patton. Please revise.

Response

Thank you for pointing it out. Cook-Patton, et al. ¹¹ did not model the belowground carbon accumulation rates using observation data. Instead, they generated a belowground carbon rate map post hoc based on their aboveground carbon sequestration rate map and IPCC default root-to-shoot ratios (see the "Mapping carbon accumulation rates" and "Data availability" sections of Cook-Patton, et al. ¹¹).

We obtained both the aboveground and belowground data directly from Susan Cook-Patton. Please find the screenshot of our email with Susan below.

We have (1) clarified how the belowground carbon sequestration rate map was generated in our Method section (Line 638-639) and (2) stated in our Data Availability section that this dataset was provided by Susan Cook-Patton (Line 767-768).

[redacted]

Comment 25

Lines 557-568: As I read this, it is assumed that tree planting can be done on abandoned cropland without first clearing the historically accumulated aboveground carbon. I think it is highly questionable to generally assume this for areas that have already been abandoned for 20-30 years. It may however be ok for newly abandoned areas or cropland that transitioned to non-forested ecosystems. It is a necessity to consider different scenarios for the fate of standing biomass on abandoned cropland at the time of active reforestation intervention. If it gets burned at site, there is an associated emission pulse involved. The authors should consult the literature to unravel the typical treatment of standing biomass on abandoned cropland and provide an assessment of its importance.

Response

Thank you for your suggestion. We have added emissions related to land clearing into our climate change mitigation potential estimation and updated all the related results. Please note that since we followed your **Comment 4(v)** to change active restoration to forest restoration via natural regrowth, we only considered emissions of land clearing in recultivation but not natural regrowth¹⁸.

Please refer to our **Response to Comment 4(v)**.

Comment 26

SI lines 235-240: The error ratios provided by Cook-Patton refers to one standard deviation of spatial variability across 100 trained machine-learning models (see Cook-Patton et al. (2020), page 9, or their online dataset description). This should be clarified better. Also, not clear to me if you did some processing to go from one standard deviation of cross-model variability to 95% confidence intervals in Supplementary Figure 5, or if this range is mis-termed as CI.

Quote Cook-Patton et al.: “Because our model is an ensemble of 100 random forest models with each random forest model trained on an independent bootstrapped sample of the training data, we can use the standard deviation of the 100 random forest models’ predictions to estimate model uncertainty in each pixel. Therefore, for each pixel, we have the model’s prediction and standard deviation across the 100 models.”

Response

Thank you for pointing it out! We have corrected the mistake on error rate and recalculated the uncertainty of our estimated climate change mitigation potential using ± 1 standard deviation (error rate) of carbon sequestration rates of natural forest regrowth (Fig. R8).

Please also see Supplementary Text 10.

Fig. R8 | Sensitivity analysis on the achievable climate change mitigation potential with spatially prioritized allocation and randomized allocation.

Possible minor typos:

Comment 27

Line 26: “pace” should be “place”.

Response

We have corrected the typo. Please see Lines 42-44.

RESPONSE TO REVIEWER #2

Comment

Key Results:

This paper provides a new map of global abandoned cropland, and evaluates how the use of that abandoned cropland for either food production or carbon sequestration by active reforestation can help reduce global hunger and help countries meet carbon emissions reductions goals. By estimating the spatial heterogeneity in both crop yields and carbon sequestration potential, the authors are able to further present an optimal spatial pattern of conversion of abandoned cropland into the two categories (food production or reforestation) to achieve the greatest combined benefit. They show that this carefully considered pattern is better than randomly allocating the land. Most importantly, they show that making use of abandoned cropland can substantially contribute to alleviating pressure on global land use.

Validity:

The authors took great care to thoroughly consider the many factors that influence the possibility of re-cultivation and reforestation. These results, along with the conservative estimates of food produced and carbon sequestered appear valid with appropriate levels of uncertainty quantification, underlying data availability, and method explanation provided.

Response

We are grateful for your insightful and encouraging comments, which helped a lot in strengthening our study. Following your suggestions, we have revised the manuscript substantially. Specifically, we have addressed methodological issues, edited texts to make our manuscript clear and informative, and performed a series of additional analyses, together with new supplementary figures, tables, data and documents. Our point-by-point responses are listed below. The corresponding revisions are marked in **red color** in the revised manuscript.

Comment 1

However, I find the conclusions regarding global food deficit reductions are not fully robust due to the lack of consideration of the global agricultural economy. In many nations, people lack sufficient calories because they cannot afford food; this issue is related but not exactly the same as insufficient food supply. I am not suggesting the authors try to include economics in their (already very detailed) analysis. Rather, I would like to see the conclusions stop at the level of calories produced or number of people who could be fed by those calories. The discussion section or supplementary text should then add a brief section on the difference between increasing food production and actually making food accessible to low-income populations.

Response

Thank you for your constructive comments. We have deleted the discussion on how the food production potential can help to reduce the global food deficit.

Instead, we followed your suggestions and added a brief section in the Discussion section

regarding (1) the difference between increasing food production and actually making food accessible to low-income populations and the difference between increasing climate change mitigation potential from abandoned cropland in a country and achieving the global-scale climate goal; and (2) what are needed to address these differences and how to put the potential from abandoned cropland into grounded benefits.

Please see Line 319-395, Line 406-418, Line 428-430, and Line 437-459.

Comment 2

The main result is a global optimization for how to use abandoned cropland. However, it is highly unlikely that all countries in the world will coordinate in a way that would enable such an optimization. The authors should address this limit in the Discussion section, as the result is valid in a hypothetical way, but not in a real-world policy recommendation way.

Response

We agree with your comment. We have added a discussion on the limitations of our result, including the hurdle of applying such globally optimized findings into real-world practice.

Please see Line 406-418, Line 428-430, and Line 437-459.

Significance:

Comment 3

These findings are significant to the field of global agricultural production, land use categorization, and the land-energy-food nexus. Perhaps of greatest significance and value to the research and policy analysis community is the data itself, specifically the geospatial data of abandoned cropland and potential yields and potential carbon sequestration on each abandoned cropland pixel. For this reason, I strongly recommend the authors post these data on a FAIR-certified public repository prior to publication.

Response

Thank you for your kind words and suggestions.

We have uploaded our data to Zenodo (a data repository suggested by Natural Portfolio journals¹ and certified by FAIRsharing²), including maps of abandoned cropland, suitable areas for recultivating and reforesting abandoned cropland, calorie food production potential and carbon sequestration potential on abandoned cropland, and simulation outcomes for key scenarios (see <https://zenodo.org/record/7758621#.ZBqPgHZByHv>).

We have signed the data availability agreement with Nature Communications and stated in the Data Availability section that we will make all these datasets freely accessible to the public prior to publication (Line 766-770). In addition, all the source data for creating Figs. 1-4 and Supplementary Data 1-5 will be also made publicly available.

¹ <https://www.nature.com/sdata/policies/repositories>

² <https://fairsharing.org/FAIRsharing.wy4egf>

Data and methodology:

Comment 4

One clarification is required: in Supplementary Data 3, Food Deficit & Supply, sources cited include links to ourworldindata.org. However, this website aggregates other publicly available data, and provides it in a GUI format. Please provide the link to the original data underlying the Our World In Data GUI. Further, the link provided for the Food Deficit data does not go to a page that includes Food Deficit. It looks like maybe the authors calculated the deficit based on the difference between minimum daily requirements and country-level calories available per capita. If this is the case, it needs to be described.

Response

Sorry for the confusion in data source. Please see the clarification below:

(1) How we calculated the annual total food deficit of each country (kcal/yr)

Annual food deficit = Depth of Food Deficit (kcal/person/day) * 365 days * Population (person)

Where the Depth of Food Deficit (kcal/person/day) in 2016 was obtained from ourworldindata.org (<https://ourworldindata.org/grapher/depth-of-the-food-deficit?tab=table>). The GUI provides the full table and downloadable .csv data; the Population of each country in 2016 was obtained from the World Bank database.

(2) How ourworldindata.org calculated the Depth of Food Deficit

The depth of the food deficit indicates how many calories would be needed to lift the undernourished from their status, everything else being constant. This indicator has been used in a book published by the World Bank

(<https://openknowledge.worldbank.org/bitstream/handle/10986/18091/9781464801334.pdf?sequence=1&openAccess=true>, pages 59-60) to provide instructions for analyzing food security.

The Depth of Food Deficit was generated by ourworldindata.org using the Suite of Food Security Indicators from FAOSTAT (<https://www.fao.org/faostat/en/#data/FS>). Ourworldindata.org offers a description of its methodology: “The average intensity of food deprivation of the undernourished, estimated as the difference between the average dietary energy requirement and the average dietary energy consumption of the undernourished population (food-deprived), is multiplied by the number of undernourished to provide an estimate of the total food deficit in the country, which is then normalized by the total population”.

Following your suggestions, we have (1) corrected the link to the data source and (2) added descriptions regarding how we calculated the country-level food deficit using the Depth of Food Deficit data and how ourworldindata.org calculated the Depth of Food Deficit, as well as the

data source.

Please see Supplementary Data 3.

Comment 5

Analytical approach:

The analytical approach is sound. I am not an expert in land cover classification methods, but the considerations the authors listed are thorough, and the sources of their methods are current.

Suggested improvements:

In addition to my suggestion about changing the results from impacting country- and global-level food deficits to just calories produced and population potentially fed by those calories, I have a few suggestions for improvements.

In Figure 3, it is important for the reader to know that the non-spatially optimized scenarios are generated with random assignment. This information needs to be mentioned earlier in the main text, before the first reference to the figure. Also, the methods text on lines 621-624 should be moved up to the main text so readers can fully understand Figure 3.

Response

Thank you for your suggestions. We have (1) changed the results to just providing how much calorie food production achievable from abandoned cropland and deleted implication analysis on addressing country-level and global-level food deficits (Line 129-133 and Supplementary Text 1); (2) moved the method description regarding the analysis on spatial prioritization to the main text (Line 251-254); (3) clearly indicated the non-spatially optimized scenarios are generated by randomized allocation before the first reference to Fig. 3 and in the caption and legend of Fig. 3 (also see Fig. R9 below)

Fig. R9 | Impact of spatial prioritization on food production potential (a) and climate change mitigation potential (b). Histograms in the upper and lower panels present the frequency distributions that summarize cropland productivity and carbon sequestration rates of abandoned cropland that are used (30% of total abandoned cropland). Lines and bars in dark and light colors indicate scenarios with employing spatial prioritization strategy and those without (i.e., randomized allocation) in using abandoned cropland, respectively. Results are showcased by three types of representative scenarios.

Comment 6

Extended Data Figure 5 is very interesting and has a significant value to the focus of the paper. This figure deserves a place in the main text, possibly more so than Figure 4 which I find has the problematic country- and global-level food security results.

Response

We have followed your suggestion and moved Extended Data Figure 5 to the main text, while deleting the country- and global-level food security results. In addition, we have provided additional analysis on (1) the possible improvements in the achievable food production potential or climate change mitigation potential on abandoned cropland and (2) how they will free up more suitable land for reforestation or recultivation, respectively. The figure for the main result is presented below.

Please see our discussion in the main text (Section - Approaches to improve the achievable potential on abandoned cropland in Line 285-361), Fig. 4, Line 732-750 (Methods) and supplementary analysis/figures/tables and related methodological details in Supplementary Text 3.

Fig. R5 | Possible approaches to improve achievable food production potential and climate change mitigation potential or to free up land for reforestation (or recultivation) while still achieving the original food production outcomes (or climate change mitigation outcomes). The error bars indicate the mean \pm standard deviation of the additional potential obtained from the freed-up abandoned cropland. Results are based on “maximizing food production” scenario and “maximizing climate change mitigation” scenario, with all suitable abandoned cropland used. Attainable yields are the 30-year mean of 2010-2040 under RCP4.5 condition (see Supplementary Text 3). PA: protected areas defined by UNEP and IUCN

Comment 7

The sentence on lines 207-209 is confusing.

Response

Sorry for the confusing description. We intended to compare the area of abandoned cropland needed to achieve the same amount of food production potential (or climate change mitigation potential) across scenarios with a spatially prioritized allocation and those without (i.e.,

randomized allocation). For example, under “food production only” scenario, to achieve 100 Pcal/yr food production potential, applying randomized allocation would require 37 Mha of abandoned cropland, while applying spatially prioritized allocation would only require 26 Mha of abandoned cropland (30% lesser than randomized allocation) (see example in Fig. R10 below). We calculated the average value of this percentage across all scenarios with spatially prioritized allocation and those without (i.e., randomized allocation). This comparison is to argue that applying spatial prioritization is an important strategy in utilizing abandoned cropland.

We have corrected the confusing description in the main text by briefly introducing how this number was calculated. Please see Line 262-268.

Fig. R10 | Illustrative diagram of the comparison between the simulated outcomes with applying a prioritized spatial allocation and those without (i.e., randomized allocation).

Comment 8

Global maps make it difficult to see details at policy-relevant scales (usually country-level or finer). It would add value to provide at least continent-level maps with country borders drawn to the Extended Data Figures section.

Response

We have provided continent-level maps in Supplementary Fig. 1 (also see below).

Fig. R10 | Abandoned cropland extent, maximum net climate change mitigation potential, and maximum food production potential breakdown for each continent.

Comment 9

Clarity and context:

The text is written clearly, and enough context is provided to understand the relevance of the study. I suggest less space be dedicated to the Tradeoffs and Synergies section, as these scenarios are all highly hypothetical and not possible due to lack of global cooperation. This would allow more space in the main text to include details currently in the Methods on a few key items that provide clarity, specifically: the methods used to identify abandoned cropland, methods for estimating managed reforestation, and the results shown in Extended Data Figure 5.

Response

Thank you for your suggestion. We have the following adjustments to each section our main text:

- **Global abandoned cropland:** We added brief descriptions regarding the methods used for cropland abandonment identification and the validations of our abandoned cropland map.
- **Food production and climate change mitigation potential:** We added brief descriptions regarding how the abandoned croplands suitable for forest restoration and recultivation were identified.
- **Trade-offs and synergies:** We moved less related context to Supplementary Texts.
- **Benefits of spatial prioritization:** We added brief descriptions to better explain spatially prioritized and non-spatially prioritized scenarios.
- ~~**Implications for food security and climate goals**~~ → We followed your suggestion to move Extended Data Fig. 5 to the main text and expanded the analysis to a new section - Approaches to improve the achievable potential on abandoned cropland (see Fig. 4, Line 285 - 361, and Supplementary Text 3)

RESPONSE TO REVIEWER #3

Comment

Dear authors, I am honored to review your work. Focusing on the food security effects and carbon sequestration potential of abandoned farmland is an interesting study. The authors assess the benefits of refarming and afforestation scenarios for abandoned farmland. The article has publication potential. However, there are still some questions to ponder before publication. Specific suggestions are as follows:

Response

We are grateful for your insightful and encouraging comments, which helped a lot in strengthening our study. Following your suggestions, we have revised the manuscript substantially. Specifically, we have addressed methodological issues, edited texts to make our manuscript clear and informative, and performed a series of additional analyses, together with new supplementary figures, tables, data and documents. Our point-by-point responses are listed below. The corresponding revisions are marked in **Blue color** in the revised manuscript.

Comment 1

How do the authors distinguish between land that can be re-farmed and land suitable for forest restoration? This information needs to be stated in detail.

Response

Sorry for not making our methodology clear. We used separate approaches to map the abandoned cropland that was suitable for recultivation (re-farmed) and/or being reforested, by considering biophysical and socioeconomic factors. It should be further noted that some abandoned cropland is suitable for both recultivation and reforestation (50% of abandoned cropland identified in our study), while others might be solely suitable for recultivation (11%) or reforestation (33%), or neither use (6%).

- **Suitable abandoned cropland for recultivation**

We used a machine learning-based approach to estimate the suitability of abandoned cropland for recultivation. We trained a maximum entropy model using driving factors of abandoned cropland recultivation and evaluated the abandoned cropland pixels that were historically recultivated (based on ESA-CCI land-cover maps). The potential drivers of recultivation covered biophysical and socioeconomic aspects, including integrated agro-ecological suitability²², market accessibility⁴², travel time to settlement⁴³, population density⁴², adjacent cropland density^{44,45}, distance to stable cropland⁴⁶, abandoned cropland density⁴⁶, and disaster-prone areas^{47,48}. We used this fine-tuned model to estimate the abandoned cropland suitable to recultivation (recultivability \in [0,1]). The recultivable abandoned cropland was determined by a recultivability larger than 0.2, while pixels with recultivability values lower than 0.2 were considered as not likely to be suitable for recultivation due to substantial biophysical, agricultural and socioeconomic barriers. We also masked out areas global protected areas to avoid adverse biological impacts. In addition, we carried out a sensitivity analysis of our approach (Supplementary Text 9).

- **Suitable abandoned cropland for reforestation**

We identified abandoned cropland suitable for reforestation with potential natural vegetation (PNV) map produced by ref. ⁴⁹. The PNV map presents areas suitable to return to native forest cover that preceded human disturbance based on biophysical, climatic, and lithological conditions, which has been widely used to identify restorable forest areas⁵⁰⁻⁵². We extracted areas suitable for forest restoration from the ‘forest’ and ‘woodland’ classes in PNV map⁵⁰ and overlaid it with our abandoned cropland map to identify abandoned cropland suitable for reforestation.

We have provided methodological details in the Method section and added a brief introduction of how we modeled and distinguished suitable areas for recultivation and reforestation in the Main text.

Please see Line 122-124, Line 139-142, Line 560-583 and Line 624-629.

Comment 2

The authors consider too few factors when considering the suitability of abandoned farmland. It has been mentioned in existing studies that disaster is one of the important driving factors affecting farmland abandonment. I suggest the authors take regional disaster as a potential driving factor for refarming.

Response

Thank you for your suggestion. In addition to the original 7 factors, we have added two disaster-related factors to improve our modeling of recultivation suitability, including INFORM Risk Index (a risk assessment index for natural and human-induced disasters produced by the European Commission)⁴⁷ and the estimated average annual economic loss due to major natural hazard (derived from the GAR Risk Atlas of the United Nations Office for Disaster Risk Reduction)⁴⁸. We also note that one of the factors that we used was agro-ecological suitability. This is an integrated indicator of whether a land is suitable for growing crops, produced by GAEZ model (FAO and IIASA)²². We have updated the other related results based on this revised outcome of recultivation suitability modeling.

Please see Line 562-567 in and Supplementary Text 7.

Comment 3

The authors refer to the carbon sequestration rate of active reforestation. In this case, active reforestation means what stage is the forest in? Seedling stage or maturity stage? The carbon sequestration rates of different growth cycles are different, which cannot be compared with the current land cover conditions. Therefore, I think the study may overestimate or underestimate the carbon sequestration potential.

Response

We agree with your point on the issue related to carbon sequestration potential estimation.

Based on one of reviewers' comment, we changed the reforestation type from active reforestation to reforestation via natural regeneration because (1) the potential adverse impact of active reforestation on biodiversity and (2) the issue related to estimating the carbon sequestration potential via active reforestation.

Hence, we changed to estimating the carbon sequestration potential on abandoned cropland by reforestation via natural regeneration. We used the latest 1-km map of carbon sequestration rates of forest natural regeneration to calculate annual climate change mitigation potential for the next 30 years¹¹. The carbon sequestration rate of this data is based on 257 historical studies and 13,112 georeferenced measurements and an ensembled machine learning model. This map provides spatially explicit carbon sequestration rates of forest natural regrowth by accounting for the variation of many factors including but not limited to climate zones, land-use conditions, stand ages, and biomes.

We agree that the factors that you mentioned are important, and will likely affect climate change mitigation potential, especially in more local-scale studies. While many of them have been considered in the carbon sequestration map that we used¹¹ (e.g., land cover conditions and biomes), some of these factors cannot yet be mapped reliably and accurately at a global scale at the moment (e.g., forest stage and local water supply condition). However, the carbon sequestration map we used is one of the best available at this moment, in terms of its spatial resolution, factors have been considered, and its relative robustness (modelling approach and huge amount of input observational data).

To account for impact of the factors that are not considered in the carbon sequestration map, we have analyzed the uncertainty of the carbon sequestration map with its error rate layer and how such uncertainty may further impact our results (see the figure below). Besides, we have added a brief discussion on the limitations of this carbon sequestration data, including the factors that are not well accounted for (e.g., differences in forest stage).

Please refer to Line 647-653, Supplementary Fig. 17, and Supplementary Text 10.

Fig. R8 | Sensitivity analysis on the achievable climate change mitigation potential with

spatially prioritized allocation and randomized allocation.

Comment 4

Line 3-4, the authors have access to the most recent data, detailed in the State of Food Security and Nutrition in the World 2022.

Response

We have updated our description on global food insecurity with the latest FAO report *the State of Food Security and Nutrition in the World 2022*. Please note that this report provides “an updated global assessment of food insecurity and nutrition for up to the year 2021”. Please see Line 16-19.

Comment 5

Line 35-37, this sentence is very confusing to me. In general, the impact of abandoned farmland on food security is negative. The authors say the link between abandoned farmland and food security is unclear, and some evidence is needed.

Response

Sorry for the confusion. We agree with you that cropland abandonment has given rise to many negative impacts on food security. However, abandoned cropland can be utilized to improve food production via recultivation and to increase carbon sequestration via reforestation. Recultivation and reforestation are the two most commonly adopted uses of abandoned cropland. The feasibility of recultivating and reforesting abandoned cropland has been proven in many studies and land-use management practices^{1,53-56}.

We have (1) provided evidence of the feasibility of using abandoned cropland to support food security (via recultivation) and climate change mitigation (via reforestation); and (2) edited Line 35-37 to improve clarity and to avoid this confusion.

Please see Line 46-55.

Comment 6

When considering the multi-use scenario, do the authors consider the use scenario that "food production can both produce carbon emissions and achieve carbon sequestration"? Abandoned farmland has an intersection in food production and climate change mitigation. For example, food production can both produce carbon emissions and achieve carbon sequestration. I suggest that the authors further analyze and study the part of "Trade-offs and synergies" to increase the depth of the article.

Response

Thank you for your constructive suggestion.

For emissions of recultivating abandoned cropland for food production, we have added the emission related to clearing historically accumulated biomass carbon for recultivation in our

net climate change mitigation potential calculation, as well as in the Trade-offs and Synergies Section.

Please see Line 133-137, Supplementary Fig. 7 and Supplementary Text 2.

There are a few approaches to creating additional carbon sequestration on cropland, including recultivated cropland. First, historically accumulated biomass before recultivation and the straw after harvest can be used for biofuel production. It has been reported that the climate change mitigation potential from this approach can not only offset the emissions from land clearing, but the life cycle emissions of crop production¹⁸. Second, nature-based climate solutions can generate additional carbon sequestration or avoid emissions. For example, “Tree in croplands” solution, “Conservation agriculture” solution, and “Improved rice cultivation” solution can offer 1.1 MgCO₂ yr⁻¹ ha⁻¹, 1.4 MgCO₂ yr⁻¹ ha⁻¹, and 1.6 MgCO_{2e} yr⁻¹ ha⁻¹, respectively, compared with the global average carbon sequestration rates of regrowing forest on suitable abandoned cropland (11.5 MgCO₂ yr⁻¹ ha⁻¹)⁵⁷. Moreover, reducing fertilizer use and improving application methods on croplands, i.e., “Cropland nutrient management” solution, can avoid N₂O emissions on cropland (706 MtCO_{2e} yr⁻¹, a rough and global scale estimate on existing cropland⁵⁷).

Nevertheless, we are not able to integrate these two approaches, i.e., biofuel production and nature-based climate solution, due to the lack of (1) spatial data (even country-level data) and (2) crop-specific data (as we considered 15 crops). We have provided discussions on how these potential approaches could bring additional climate benefits to the recultivated abandoned cropland.

Please see Line 349-361.

Reference

- 1 Crawford, C. L., Yin, H., Radeloff, V. C. & Wilcove, D. S. Rural land abandonment is too ephemeral to provide major benefits for biodiversity and climate. *Science Advances* **8**, eabm8999, doi:doi:10.1126/sciadv.abm8999 (2022).
- 2 Weng, Q. *Scale issues in remote sensing*. (John Wiley & Sons, 2014).
- 3 Lu, M. *et al.* A comparative analysis of five global cropland datasets in China. *Science China Earth Sciences* **59**, 2307-2317 (2016).
- 4 Næss, J. S., Cavalett, O. & Cherubini, F. The land–energy–water nexus of global bioenergy potentials from abandoned cropland. *Nature Sustainability* **4**, 525-536, doi:10.1038/s41893-020-00680-5 (2021).
- 5 Dawe, D., Jaffee, S. & Santos, N. Rice in the shadow of skyscrapers: Policy choices in a dynamic East and Southeast Asian setting. (2014).
- 6 Olsen, V. M. *et al.* The impact of conflict-driven cropland abandonment on food insecurity in South Sudan revealed using satellite remote sensing. *Nature Food* **2**, 990-996, doi:10.1038/s43016-021-00417-3 (2021).
- 7 Comber, A., Fisher, P. & Wadsworth, R. Actor–network theory: a suitable framework to understand how land cover mapping projects develop? *Land Use Policy* **20**, 299-309 (2003).

- 8 Comber, A., Fisher, P. & Wadsworth, R. Integrating land-cover data with different ontologies: identifying change from inconsistency. *International Journal of Geographical Information Science* **18**, 691-708 (2004).
- 9 Defourny, P. *et al.* Land cover CCI. *Product User Guide Version 2*, 325 (2012).
- 10 Yin, H. *et al.* Monitoring cropland abandonment with Landsat time series. *Remote Sensing of Environment* **246**, doi:10.1016/j.rse.2020.111873 (2020).
- 11 Cook-Patton, S. C. *et al.* Mapping carbon accumulation potential from global natural forest regrowth. *Nature* **585**, 545-550, doi:10.1038/s41586-020-2686-x (2020).
- 12 Spawn, S. A., Sullivan, C. C., Lark, T. J. & Gibbs, H. K. Harmonized global maps of above and belowground biomass carbon density in the year 2010. *Scientific data* **7**, 112, doi:10.1038/s41597-020-0444-4 (2020).
- 13 Perrin, A., Basset-Mens, C. & Gabrielle, B. Life cycle assessment of vegetable products: a review focusing on cropping systems diversity and the estimation of field emissions. *The International Journal of Life Cycle Assessment* **19**, 1247-1263 (2014).
- 14 Riahi, K. *et al.* The Shared Socioeconomic Pathways and their energy, land use, and greenhouse gas emissions implications: An overview. *Global Environmental Change* **42**, 153-168, doi:10.1016/j.gloenvcha.2016.05.009 (2017).
- 15 IUCN, U.-W. a. *Protected Planet: The World Database on Protected Areas (WDPA)*, <www.protectedplanet.net> (2022).
- 16 Fradette, O. *et al.* Additional carbon sequestration potential of abandoned agricultural land afforestation in the boreal zone: A modelling approach. *Forest Ecology and Management* **499**, doi:10.1016/j.foreco.2021.119565 (2021).
- 17 Bell, S. M., Barriocanal, C., Terrer, C. & Rosell-Melé, A. Management opportunities for soil carbon sequestration following agricultural land abandonment. *Environmental Science & Policy* **108**, 104-111, doi:10.1016/j.envsci.2020.03.018 (2020).
- 18 Gvein, M. H. *et al.* Potential of land-based climate change mitigation strategies on abandoned cropland. *Communications Earth & Environment* **4**, doi:10.1038/s43247-023-00696-7 (2023).
- 19 Sadras, V. *et al.* Yield gap analysis of field crops: Methods and case studies. (2015).
- 20 Díaz, G. I., Nahuelhual, L., Echeverría, C. & Marín, S. Drivers of land abandonment in Southern Chile and implications for landscape planning. *Landscape and Urban Planning* **99**, 207-217, doi:10.1016/j.landurbplan.2010.11.005 (2011).
- 21 Lasanta, T. *et al.* Space-time process and drivers of land abandonment in Europe. *Catena* **149**, 810-823, doi:10.1016/j.catena.2016.02.024 (2017).
- 22 Fischer, G. *et al.* Global Agro-ecological Zones (GAEZ v4)-Model Documentation. (2021).
- 23 Rattalino Edreira, J. I. *et al.* Spatial frameworks for robust estimation of yield gaps. *Nature Food* **2**, 773-779, doi:10.1038/s43016-021-00365-y (2021).
- 24 Grogan, D., Frohling, S., Wisser, D., Prusevich, A. & Glidden, S. Global gridded crop harvested area, production, yield, and monthly physical area data circa 2015. *Scientific data* **9**, 15, doi:10.1038/s41597-021-01115-2 (2022).
- 25 Rosenzweig, C. *et al.* Assessing agricultural risks of climate change in the 21st century in a global gridded crop model intercomparison. *Proceedings of the National Academy of Sciences of the United States of America* **111**, 3268-3273, doi:10.1073/pnas.1222463110 (2014).
- 26 Pu, L., Zhang, S., Yang, J., Chang, L. & Bai, S. Spatio-Temporal Dynamics of Maize Potential Yield and Yield Gaps in Northeast China from 1990 to 2015. *Int J Environ Res Public Health*

- 16, doi:10.3390/ijerph16071211 (2019).
- 27 Deng, N. *et al.* Closing yield gaps for rice self-sufficiency in China. *Nat Commun* **10**, 1725, doi:10.1038/s41467-019-09447-9 (2019).
- 28 Rosenzweig, C. *et al.* Assessing agricultural risks of climate change in the 21st century in a global gridded crop model intercomparison. *Proceedings of the national academy of sciences* **111**, 3268-3273 (2014).
- 29 Teixeira, E. I., Fischer, G., Van Velthuisen, H., Walter, C. & Ewert, F. Global hot-spots of heat stress on agricultural crops due to climate change. *Agricultural and Forest Meteorology* **170**, 206-215 (2013).
- 30 Mauser, W. *et al.* Global biomass production potentials exceed expected future demand without the need for cropland expansion. *Nature communications* **6**, 8946 (2015).
- 31 Olofsson, P., Foody, G. M., Stehman, S. V. & Woodcock, C. E. Making better use of accuracy data in land change studies: Estimating accuracy and area and quantifying uncertainty using stratified estimation. *Remote Sensing of Environment* **129**, 122-131, doi:10.1016/j.rse.2012.10.031 (2013).
- 32 Olofsson, P. *et al.* Good practices for estimating area and assessing accuracy of land change. *Remote Sensing of Environment* **148**, 42-57, doi:10.1016/j.rse.2014.02.015 (2014).
- 33 Yin, H. *et al.* Mapping agricultural land abandonment from spatial and temporal segmentation of Landsat time series. *Remote Sensing of Environment* **210**, 12-24, doi:10.1016/j.rse.2018.02.050 (2018).
- 34 Wyborn, C. & Evans, M. C. Conservation needs to break free from global priority mapping. *Nat Ecol Evol* **5**, 1322-1324, doi:10.1038/s41559-021-01540-x (2021).
- 35 Chaplin-Kramer, R. *et al.* Conservation needs to integrate knowledge across scales. *Nat Ecol Evol* **6**, 118-119, doi:10.1038/s41559-021-01605-x (2022).
- 36 Filoso, S., Bezerra, M. O., Weiss, K. C. B. & Palmer, M. A. Impacts of forest restoration on water yield: A systematic review. *PLoS One* **12**, e0183210, doi:10.1371/journal.pone.0183210 (2017).
- 37 Feng, X. *et al.* Revegetation in China's Loess Plateau is approaching sustainable water resource limits. *Nature Climate Change* **6**, 1019-1022, doi:10.1038/nclimate3092 (2016).
- 38 Teo, H. C. *et al.* Large-scale reforestation can increase water yield and reduce drought risk for water-insecure regions in the Asia-Pacific. *Glob Chang Biol*, doi:10.1111/gcb.16404 (2022).
- 39 Betts, R. A. Offset of the potential carbon sink from boreal forestation by decreases in surface albedo. *Nature* **408**, 187-190, doi:10.1038/35041545 (2000).
- 40 Clark, M. L., Aide, T. M., Grau, H. R. & Riner, G. A scalable approach to mapping annual land cover at 250 m using MODIS time series data: A case study in the Dry Chaco ecoregion of South America. *Remote Sensing of Environment* **114**, 2816-2832 (2010).
- 41 Zhang, L. & Weng, Q. Annual dynamics of impervious surface in the Pearl River Delta, China, from 1988 to 2013, using time series Landsat imagery. *ISPRS Journal of Photogrammetry and Remote Sensing* **113**, 86-96, doi:10.1016/j.isprsjprs.2016.01.003 (2016).
- 42 Estel, S. *et al.* Mapping farmland abandonment and recultivation across Europe using MODIS NDVI time series. *Remote Sensing of Environment* **163**, 312-325, doi:10.1016/j.rse.2015.03.028 (2015).
- 43 Zumkehr, A. & Campbell, J. E. Historical U.S. cropland areas and the potential for bioenergy production on abandoned croplands. *Environmental science & technology* **47**, 3840-3847,

- doi:10.1021/es3033132 (2013).
- 44 Zhao, H. *et al.* China's future food demand and its implications for trade and environment. *Nature Sustainability* **4**, 1042-1051, doi:10.1038/s41893-021-00784-6 (2021).
- 45 Dara, A. *et al.* Mapping the timing of cropland abandonment and recultivation in northern Kazakhstan using annual Landsat time series. *Remote Sensing of Environment* **213**, 49-60, doi:10.1016/j.rse.2018.05.005 (2018).
- 46 Meyfroidt, P., Schierhorn, F., Prishchepov, A. V., Müller, D. & Kuemmerle, T. Drivers, constraints and trade-offs associated with recultivating abandoned cropland in Russia, Ukraine and Kazakhstan. *Global Environmental Change* **37**, 1-15, doi:10.1016/j.gloenvcha.2016.01.003 (2016).
- 47 De Groeve, T., Poljansek, K. & Vernaccini, L. Index for risk management-INFORM. *JRC Science for Policy Reports (Brussels: European Commission)* (2015).
- 48 Reduction, U. N. O. f. D. R. (UNISDR Geneva, Switzerland, 2017).
- 49 Hengl, T. *et al.* Global mapping of potential natural vegetation: an assessment of machine learning algorithms for estimating land potential. *PeerJ* **6**, e5457, doi:10.7717/peerj.5457 (2018).
- 50 Zeng, Y. *et al.* Economic and social constraints on reforestation for climate mitigation in Southeast Asia. *Nature Climate Change* **10**, 842-844, doi:10.1038/s41558-020-0856-3 (2020).
- 51 Chiarucci, A., Araújo, M. B., Decocq, G., Beierkuhnlein, C. & Fernández-Palacios, J. M. The concept of potential natural vegetation: an epitaph? *Journal of Vegetation Science* **21**, 1172-1178 (2010).
- 52 Török, K. *et al.* Restoration prioritization for industrial area applying multiple potential natural vegetation modeling. *Restoration Ecology* **26**, 476-488 (2018).
- 53 Yang, Y. *et al.* Restoring Abandoned Farmland to Mitigate Climate Change on a Full Earth. *One Earth* **3**, 176-186, doi:10.1016/j.oneear.2020.07.019 (2020).
- 54 Chapman, C. A. & Chapman, L. J. Forest restoration in abandoned agricultural land: a case study from East Africa. *Conservation Biology* **13**, 1301-1311 (1999).
- 55 Schulte, L. A. *et al.* Meeting global challenges with regenerative agriculture producing food and energy. *Nature Sustainability*, doi:10.1038/s41893-021-00827-y (2021).
- 56 Tomaz, C., Alegria, C., Monteiro, J. M. & Teixeira, M. C. Land cover change and afforestation of marginal and abandoned agricultural land: A 10year analysis in a Mediterranean region. *Forest Ecology and Management* **308**, 40-49, doi:10.1016/j.foreco.2013.07.044 (2013).
- 57 Griscom, B. W. *et al.* Natural climate solutions. *Proceedings of the National Academy of Sciences of the United States of America* **114**, 11645-11650, doi:10.1073/pnas.1710465114 (2017).

Reviewers' Comments:

Reviewer #1:

Remarks to the Author:

Dear Authors,

This revision effort has strengthened the manuscript substantially. I am now increasingly convinced that the study merits a high-impact publication. It has become more robust, some key previous weaknesses have been eliminated, and I find the remaining limitations mostly approaching acceptable for a broad global study such as this. The study now presents an extensive and impressive prioritization analysis of two land management options for abandoned cropland. I have a few more comments for consideration, including one major point.

Best regards,
Jan Sandstad Næss

Main comments:

One key question I have now, is what role the "single-purpose food production only" scenario currently plays. In this case, 61 Mha is allocated to food production only, whilst the remaining abandoned cropland remains unmanaged. As I understand, the remaining 40 Mha deemed non-recultivable by the maximum entropy model is therefore predominantly subject to continued natural regrowth (33 Mha were found suitable for reforestation). This indicates that the land is also serving two purposes here, and this scenario seems in practice equal to the "multi-purpose maximizing food production scenario", but without getting credited for regrowth. I am wondering if the "single-purpose food production only" scenario has now become redundant with revised methodology.

Specific in-line comments:

Abstract, line 5-6. It may be good to explain very briefly what lies behind the ranges presented here (calories/mitigation), e.g., that it is across scenarios/depends on management strategies.

Line 22. Might be fair to cite the SSP overview paper here also at first mention, although I see you have done that later in the manuscript (e.g., Riahi et al. (2017)).

Riahi, Keywan, et al. "The Shared Socioeconomic Pathways and their energy, land use, and greenhouse gas emissions implications: An overview." *Global environmental change* 42 (2017): 153-168. <https://doi.org/10.1016/j.gloenvcha.2016.05.009>

Lines 46-47. Not completely sure why "biodiversity and carbon stocks" is threatened by cropland abandonment. Should these words be deleted? The sentence would make more sense then.

Lines 53-55. The first part of this sentence sounds a bit of a "heavy read" to me. The content is ok, but the sentence may benefit from a rewrite.

Line 137. I suggest to also provide the total carbon emissions that happens instantly here and not only the 30-year average (about 4.7Gt CO₂?).

Line 294 (Fig4). Agree with moving this one to the main text!

Lines 344-349. I think you made the right call to rely directly on Cook-Patton for the spatial prioritization analysis. The aggregate active afforestation estimate has now become a useful quantification that strengthens the study. Could it be suitable to include the standard deviation from Supplementary Data 5 here too?

Lines 382-384. Basically, you have now shown that predicted GAEZ rain-fed food crop yields on the abandoned cropland on average increases marginally with increased global warming (e.g., Supplementary Table 3), which does not completely match this general statement based on Xu et al. Can this be attributed to the geospatial distribution of the abandoned cropland?

Previous studies using a variety of models have shown that predicted yields generally increase at higher latitudes and decrease at lower latitudes with increased warming (for example, Rosenzweig et al.'s evaluation of four crops that is already cited). There is also some cross-model variation, which can contribute to divergencies. If it is attributable to the land distribution, then this may be worth a mention either in the main text or SI.

Lines 428-446. This is a very nice discussion, appreciate the nuances of local-global context.

Lines 520-525. Provide some citations to papers evaluating these limitations and to Landsat and Sentinel?

Lines 589-593. Overall, you describe some validation efforts here, which helps build confidence in GAEZ v4 outputs. However, the text doesn't really reveal the scientific insights obtained from these previous studies. Also, previous investigations are somewhat limited in their extent. The validation and comparison exercises were done for different versions of the GAEZ model (e.g., GAEZ-IMAGE, GAEZ 3, GAEZ+ (which builds on GAEZ4) ...). Rosenzweig et al. and Grogan et al. addresses four of the fifteen crops (maize, rice, soy, and wheat). Pu et al. addressed Northeast China only. Rattalino Edreira et al. focused on maize, rice, and wheat. Deng et al. on rice in China.

It seems important to pass on how GAEZ positions itself relative to these other models and whether there are any consistent biases relative to observations. Also, what crops have been investigated and which ones have not. I think enough space is allocated in the main text for this topic, but it could be useful to go a bit more in-depth in a supplementary text.

Lines 599-601. Both observed and predicted yields in specific years can be sensitive to above-average or below-average temperatures, precipitation etc. Were these years (2009-2011) representative of the average 30-year climate or especially dry/wet/warm/cold? If they diverged from the mean this should be mentioned and potential implications discussed (maybe linked to the previous comment). ECMWF typically provides some analysis, but not sure how far back in time it goes (<https://climate.copernicus.eu/esotc/2021>).

Line 601. Spawn et al. does not address GAEZ. I think this citation might be misplaced.

Lines 666-669. I acknowledge the point made that it is a challenge to provide robust geospatial estimates of life-cycle emissions of food production. It still seems beneficial to provide a literature estimate/example of on-farm and supply chain emissions even if this is not included in the prioritization (especially a high-end estimate with major use of fertilizers and mechanization). Ecoinvent could also be checked.

Line 743. Should include some citations to RCP45 and RCP85. E.g., Thomson et al. (2011) and Riahi et al. (2011). Also, it is worth mentioning briefly in the main text also that you considered mean yields based on multiple climate models and to consider providing a cluster citation to these individual climate models.

Thomson, A.M., Calvin, K.V., Smith, S.J. et al. RCP4.5: a pathway for stabilization of radiative forcing by 2100. *Climatic Change* 109, 77 (2011). <https://doi.org/10.1007/s10584-011-0151-4>

Riahi, K., Rao, S., Krey, V. et al. RCP 8.5—A scenario of comparatively high greenhouse gas emissions. *Climatic Change* 109, 33 (2011). <https://doi.org/10.1007/s10584-011-0149-y>

Lines 767-768. Thank you for this clarification, happy to be proven wrong!

Supplementary Tables 5-6. This is highly valuable, and it was about time that this validation exercise was done (it should have been done also by previous studies). I now feel that choice of sample size is better justified. However, I notice that producer's and user's accuracies seem inconsistent across the two tables. In Supplementary Table 5, producer accuracy is 0.770 and user accuracy is 0.934 (this aligns with the main text), whilst in Supplementary Table 6 continental producer accuracies range between 0.84-0.95 and user accuracies range between 0.71-0.80. Is it possible that the column headers "Producer's accuracy" and "User's accuracy" have been switched in Supplementary Table 6?

Supplementary text 3. Lines 119-120. Seems like these climate models should be cited here.

Supplementary text 6. Line 325. Supplementary Data 5 is the dataset with active afforestation conversion factors in the files I have access to. Should this reference be to Supplementary Data 1? Need to double check for similar possible cross-reference errors in the manuscript and SI.

Supplementary Text 8. This is interesting, the discussion complements the previous analysis done by Crawford et al. and highlights how different methods may lead to varying conclusions. A key takeaway may be that we do not yet fully understand the historical importance of the recultivation phenomenon. I would appreciate some more clarifications, as follows.

SI lines 383-384. Both Crawford et al. and Yin et al. consider a non-vegetated class that includes built-up areas. It was claimed that cropland conversion to non-vegetated classes were excluded from the abandonment mapping (see p10, Crawford et al.). Might the overlap seen in Supplementary Fig 14 therefore also be linked to the accuracy of this non-vegetated class?

What about the 1-year recultivation threshold, can this increase the risk of single-year pixel misclassifications affecting conclusions and contribute to the seen differences?

SI lines 390-400. I totally agree with this. It is mostly about having an open discussion that takes us closer to our common goal of understanding abandonment, recultivation, and land use change. And these previous studies are important.

SI lines 400-405. I don't think that Supplementary Fig15 completely supports the conclusion you make here. A map of abandoned cropland extent in hectare (Supplementary Fig15a) is compared to locational proportions of abandoned land recultivated by 2017 (Supplementary Fig15b). Shouldn't hectares be compared with hectares and proportional recultivation with proportional recultivation? Of course, whilst clearly referring to the methodological differences in evaluated period, resolution etc. as already done.

Supplementary Text 8. The mapped historical recultivation seems like a key input to the approach that is used to quantify recultivation suitability. Based on the highlighted importance of methodological choices on quantified recultivation, I was hoping that you could reflect on how the use of a different method/dataset might in turn affect your results. Would it be likely to have a major impact, or can it be expected that the main conclusions would not change?

Potential minor typos:

Line 342. "expereience".

SI line 91. "when model when modelling[...]".

Thank you for making it easy to spot changes through the color codes. Also appreciated the willingness to make data openly available after publication.

Reviewer #2:

Remarks to the Author:

I find that the authors adequately addressed all of my concerns. The added clarity around methods and figures leaves me with no remaining questions. Thank you for taking all of my comments into account, and especially for adjusting the conclusions regarding food provisioning.

Reviewer #3:

Remarks to the Author:

Thanks to the authors for their efforts. I am satisfied with the revision and recommend that it be published in its current form.

RESPONSE TO REVIEWER #1

General comment

Dear Authors,

This revision effort has strengthened the manuscript substantially. I am now increasingly convinced that the study merits a high-impact publication. It has become more robust, some key previous weaknesses have been eliminated, and I find the remaining limitations mostly approaching acceptable for a broad global study such as this. The study now presents an extensive and impressive prioritization analysis of two land management options for abandoned cropland. I have a few more comments for consideration, including one major point.

Thank you for making it easy to spot changes through the color codes. Also appreciated the willingness to make data openly available after publication.

Best regards,

Jan Sandstad Næss

Response

Dear Dr. Jan Sandstad Næss,

We are grateful for your insightful and encouraging comments, which helped us to further improve our manuscript in Round 2 revision. Following your suggestions, we have addressed the major concern (redundant single-purpose scenario), as well as other minor issues, by performing additional analyses, adding new supplementary texts & figures, and editing text. Our point-by-point responses are listed below. The corresponding revisions are marked in **green color** in the revised manuscript.

Best regards,

Qiming Zheng, on behalf of the authors.

Comment #1

One key question I have now, is what role the “single-purpose food production only” scenario currently plays. In this case, 61 Mha is allocated to food production only, whilst the remaining abandoned cropland remains unmanaged. As I understand, the remaining 40 Mha deemed non-recultivable by the maximum entropy model is therefore predominantly subject to continued natural regrowth (33 Mha were found suitable for reforestation). This indicates that the land is also serving two purposes here, and this scenario seems in practice equal to the “multi-purpose maximizing food production scenario”, but without getting credited for regrowth. I am wondering if the “single-purpose food production only” scenario has now become redundant with revised

methodology.

Response

Thank you so much for your comments. We understand your perspective and agree that the “single-purpose” scenarios are now somewhat redundant. Also, most of our analyses focus on discussing the trade-offs and synergies across “multi-purpose scenarios”. Following your suggestion, we now remove all the content related to “single-purpose scenarios”.

Comment #2

Abstract, line 5-6. It may be good to explain very briefly what lies behind the ranges presented here (calories/mitigation), e.g., that it is across scenarios/depends on management strategies.

Response

Following your suggestion, we added a brief explanation of how the ranges of calories/mitigation were derived – depending on land-use suitability and different land allocation strategies. We are, however, still constrained by the 150-word limit (now 149 words).

Please see Line 5-9.

Comment #3

Line 22. Might be fair to cite the SSP overview paper here also at first mention, although I see you have done that later in the manuscript (e.g., Riahi et al. (2017)).

Riahi, Keywan, et al. "The Shared Socioeconomic Pathways and their energy, land use, and greenhouse gas emissions implications: An overview." *Global environmental change* 42 (2017): 153-168. <https://doi.org/10.1016/j.gloenvcha.2016.05.009>

Response

Thank you and we added the suggested reference accordingly. Please also see Line 22-25.

Comment #4

Lines 46-47. Not completely sure why “biodiversity and carbon stocks” is threatened by cropland abandonment. Should these words be deleted? The sentence would make more sense then.

Response

Thanks for pointing that out. We agree and deleted these words as suggested. Please also see Line 47-51.

Comment #5

Lines 53-55. The first part of this sentence sounds a bit of a “heavy read” to me. The content is ok, but the sentence may benefit from a rewrite.

Response

Thank you for your suggestion. We rewrote the sentence as follows:

“It remains unclear regarding how recultivating and reforesting global abandoned cropland can support food production and climate change mitigation, respectively, and how they are pertinent to

achieve food security and climate goals.”

Please also see Line 53-55.

Comment #6

Line 137. I suggest to also provide the total carbon emissions that happens instantly here and not only the 30-year average (about 4.7Gt CO₂?).

Response

Thank you for your comment. We added the total emissions incurred by land clearing together with the 30-year average. Please also see Line 134-136.

Comment #7

Line 294 (Fig4). Agree with moving this one to the main text!

Response

Thank you for your encouragement!

Comment #8

Lines 344-349. I think you made the right call to rely directly on Cook-Patton for the spatial prioritization analysis. The aggregate active afforestation estimate has now become a useful quantification that strengthens the study. Could it be suitable to include the standard deviation from Supplementary Data 5 here too?

Response

Following your suggestion, we used the mean and standard deviation of the conversion factor in Supplementary Data 5 to provide the mean+std increase (53±18%) in the climate change mitigation potential if we actively reforest abandoned cropland outside global protected areas.

Please also see Line 337-340.

Comment #9

Lines 382-384. Basically, you have now shown that predicted GAEZ rain-fed food crop yields on the abandoned cropland on average increase marginally with increased global warming (e.g., Supplementary Table 3), which does not completely match this general statement based on Xu et al.

Can this be attributed to the geospatial distribution of the abandoned cropland?

Previous studies using a variety of models have shown that predicted yields generally increase at higher latitudes and decrease at lower latitudes with increased warming (for example, Rosenzweig et al.'s evaluation of four crops that is already cited). There is also some cross-model variation, which can contribute to divergencies. If it is attributable to the land distribution, then this may be

worth a mention either in the main text or SI.

Response

Thank you for your comments. First, our key purpose of citing Xu, et al. ¹ is to highlight the importance of deploying bioenergy crops as another option of re-using abandoned cropland rather than to discuss about crop yields.

Second, the mismatch in future crop yields between our study (marginally increase) and Xu, et al. ¹ (decrease) could be ascribed to several reasons.

1. **Geospatial distribution.** Xu, et al. ¹ looked at global cropland, while we only focused on global abandoned cropland.
2. **Time frame.** Xu, et al. ¹ (2020-2200) and Rosenzweig, et al. ² (1980-2100) mainly focused on predicting long-term yield changes, while we focused on medium-term yield changes (i.e., 2010-2040 and 2040-2070). The difference lies also in future temperature increases. We expect a much higher temperature increase at the end of the century (RCP4.5: +1.8°C; RCP8.5: +3.7°C) than during the mid of the century (RCP4.5: +1.4°C; RCP8.5: +2.0°C)^a. Therefore, the negative impact of an increased temperature and precipitation shortfall on crop yields is more severe in Xu, et al. ¹ and Rosenzweig, et al. ² (long-term crop yield projections) (also see Fig. 4 of Rosenzweig, et al. ² below).

[redacted]

Figure R1. Relative changes (%) of crop yields under RCP8.5. This figure is adopted from Fig. 4 of Rosenzweig, et al. ².

3. **Yield type.** Our study used the future attainable yields from GAEZ v4, i.e., yields under optimal management practices (e.g., non-limiting nutrient condition, effective control of biotic stresses, etc.). Xu, et al. ¹ relied on modeled actual yields. Some models in Rosenzweig, et al. ² simulated actual yields, while others simulated potential yields (see caption of Fig. 2 of Rosenzweig, et al. ²).

^a IPCC AR5: <https://web.archive.org/web/20181223030119/https://archive.ipcc.ch/report/ar5/wg1/>

4. **Crop types.** Our study was based on harvest area weighted yields of 15 crops. Different crops have different responses to future climate change (Figure R2).

Figure R2. The difference in rain-fed attainable yields of four crops used in our study. Data source: GAEZ v4.

Regardless, we understand this concern and added a discussion on the future attainable crop yield data used in our study and how future crop yields could be affected by global warming. Please see **Supplementary Text 3 – Discussion 1**.

Comment #10

Lines 428-446. This is a very nice discussion, appreciate the nuances of local-global context.

Response

Thank you for your encouraging words.

Comment #11

Lines 520-525. Provide some citations to papers evaluating these limitations and to Landsat and Sentinel?

Response

Thank you for your suggestion. We added references about (1) limitations of coarse resolution land cover products (or cropland map products)³⁻⁵ and (2) how Landsat and Sentinel data can help to improve cropland abandonment mapping^{6,7}.

Please also see Line 515-522.

Comment #12

Lines 589-593. Overall, you describe some validation efforts here, which helps build confidence in GAEZ v4 outputs. However, the text doesn't really reveal the scientific insights obtained from these previous studies. Also, previous investigations are somewhat limited in their extent. The validation and comparison exercises were done for different versions of the GAEZ model (e.g., GAEZ-IMAGE, GAEZ 3, GAEZ+ (which builds on GAEZ4) ...). Rosenzweig et al. and Grogan et al. addresses four of the fifteen crops (maize, rice, soy, and wheat). Pu et al. addressed Northeast China only. Rattalino Edreira et al. focused on maize, rice, and wheat. Deng et al. on rice in China.

It seems important to pass on how GAEZ positions itself relative to these other models and whether there are any consistent biases relative to observations. Also, what crops have been investigated and which ones have not. I think enough space is allocated in the main text for this topic, but it could be useful to go a bit more in-depth in a supplementary text.

Response

Thank you for your suggestion. We added a brief discussion in **Supplementary Text 3 – Discussion 2** to explain the limitations and biases in existing validation efforts of GAEZ crop yield data in light of crop types, GAEZ model versions, and study areas.

Comment #13

Lines 599-601. Both observed and predicted yields in specific years can be sensitive to above-average or below-average temperatures, precipitation etc. Were these years (2009-2011) representative of the average 30-year climate or especially dry/wet/warm/cold? If they diverged from the mean this should be mentioned and potential implications discussed (maybe linked to the previous comment). ECMWF typically provides some analysis, but not sure how far back in time it goes (<https://climate.copernicus.eu/esotc/2021>).

Response

Thank you for your insightful suggestion. We checked the global average temperature time series^b and precipitation anomaly time series^c. We found that present-day actual yield data that we obtained from GAEZ v4 (2009-2011) are at years of normal global temperature but at years with more precipitations. This may lead to a higher actual yield of 2009-2011 than the 30-year average level.

We have added a brief discussion on the fact that the actual yield data are at relatively wetter years and may be higher than the 30-year average condition. Please see Line 595-601.

^b <https://climate.copernicus.eu/esotc/2021/globe-in-2021>

^c <https://www.epa.gov/climate-indicators/climate-change-indicators-us-and-global-precipitation>

[redacted]

Figure R3. Global temperature time series and global precipitation anomaly time series. Figures are adopted from Copernicus Climate Change Service Climate indicator and US EPA, respectively.

Comment #14

Line 601. Spawn et al. does not address GAEZ. I think this citation might be misplaced.

Response

We have deleted this misplaced citation. Thank you for pointing it out.

Comment #15

Lines 666-669. I acknowledge the point made that it is a challenge to provide robust geospatial estimates of life-cycle emissions of food production. It still seems beneficial to provide a literature estimate/example of on-farm and supply chain emissions even if this is not included in the prioritization (especially a high-end estimate with major use of fertilizers and mechanization). Ecoinvent could also be checked.

Response

Thank you for your insightful suggestion. We added an example of life cycle emission of wheat production in Western Australia (308 kgCO₂e/t)⁸. It accounts for the emissions of prefarm stage (136 kgCO₂e/t; agricultural machinery, fertilizer and pesticide production), onfarm stage (135 kgCO₂e/t; diesel use, liming and nitrous oxide emissions), and postfarm stage (34 kgCO₂e/t; grain storage and transportation to the port).

By providing this example, we believe it could give the readers a broad sense of the life cycle emission of recultivating abandoned cropland.

Please also see Line 667-672.

Comment #16

Line 743. Should include some citations to RCP45 and RCP85. E.g., Thomson et al. (2011) and Riahi et al. (2011). Also, it is worth mentioning briefly in the main text also that you considered mean yields based on multiple climate models and to consider providing a cluster citation to these individual climate models.

Thomson, A.M., Calvin, K.V., Smith, S.J. et al. RCP4.5:. Climatic Change 109, 77 (2011). <https://doi.org/10.1007/s10584-011-0151-4>

Riahi, K., Rao, S., Krey, V. et al. RCP 8.5—s. Climatic Change 109, 33 (2011). <https://doi.org/10.1007/s10584-011-0149-y>

Response

Following your suggestion, we added two suggested references of RCP45 and RCP85 and explained that our study considered mean yields based on multiple climate models.

Based on GAEZ v4 document⁹, the future climate forcing was the ensemble mean of five climate models GFDL-ESM2M, HadGEM2-ES, IPSL-CM5A-LR, MIROC-ESM-CHEM, and NorESM1-M, obtained from bias-corrected the Intersectoral Impact Model Intercomparison Project (ISI-MIP)¹⁰. Instead of providing reference for each climate model, we added the ref. ¹⁰ that is also used in the GAEZ v4 document p15.

Please also see Line 738-739 and Line 739-742.

Comment #17

Lines 767-768. Thank you for this clarification, happy to be proven wrong!

Response

You are welcome, and we appreciate your kind words.

Comment #18

Supplementary Tables 5-6. This is highly valuable, and it was about time that this validation exercise was done (it should have been done also by previous studies). I now feel that choice of sample size is better justified. However, I notice that producer's and user's accuracies seem inconsistent across the two tables. In Supplementary Table 5, producer accuracy is 0.770 and user accuracy is 0.934 (this aligns with the main text), whilst in Supplementary Table 6 continental producer accuracies range between 0.84-0.95 and user accuracies range between 0.71-0.80. Is it possible that the column headers "Producer's accuracy" and "User's accuracy" have been switched in Supplementary Table 6?

Response

Thanks for pointing this out. We agree and corrected the order of column headers of Supplementary Table 6.

Comment #19

Supplementary text 3. Lines 119-120. Seems like these climate models should be cited here.

Response

Thank you for your suggestion. Based on GAEZ v4 document⁹, the future climate forcing was the ensemble mean of five climate models GFDL-ESM2M, HadGEM2-ES, IPSL-CM5A-LR, MIROC-ESM-CHEM, and NorESM1-M, which were obtained from bias-corrected the Intersectoral Impact Model Intercomparison Project (ISI-MIP)¹⁰. Instead of providing reference for each climate model, we added the ref. ¹⁰ that is also used in the GAEZ v4 document p15.

Please also see Line 739-742.

Comment #20

Supplementary text 6. Line 325. Supplementary Data 5 is the dataset with active afforestation conversion factors in the files I have access to. Should this reference be to Supplementary Data 1? Need to double check for similar possible cross-reference errors in the manuscript and SI.
?

Response

Thank you for pointing it out. It should be "Supplementary Data 1". We have corrected this error, as well as other similar cross-reference errors.

Comment #21

Supplementary Text 8. This is interesting, the discussion complements the previous analysis done by Crawford et al. and highlights how different methods may lead to varying conclusions. A key takeaway may be that we do not yet fully understand the historical importance of the recultivation phenomenon. I would appreciate some more clarifications, as follows.

SI lines 383-384. Both Crawford et al. and Yin et al. consider a non-vegetated class that includes built-up areas. It was claimed that cropland conversion to non-vegetated classes were excluded from the abandonment mapping (see p10, Crawford et al.). Might the overlap seen in Supplementary Fig 14 therefore also be linked to the accuracy of this non-vegetated class?

Response

Thank you for your comments.

Even though Crawford, et al. ¹¹ stated that “pixels that transitioned from cropland to non-vegetated classes were not considered as abandoned and were excluded”, we still found some pixels transitioned to road/settlement were labeled as abandoned cropland based on the source abandoned cropland map that Crawford, et al. ¹¹ shared (<https://zenodo.org/record/5348287#.ZBiOXXZByUn>). However, we are not able to tell whether these non-vegetated classes affect the accuracy of their abandoned cropland and recultivation maps.

Nevertheless, after a careful consideration, we think it would be better to delete Supplementary Fig. 14 based on two main reasons. (1) The key purpose of Supplementary Text 8 is to discuss the possible reasons for a different recultivation pattern obtained from our study and Crawford, et al. ¹¹. We have already discussed these reasons from the perspective of the differences in spatial resolution, study areas and definition of abandonment pixels in Supplementary Text 8. In addition, we have also validated our abandonment map and provided sampling-based approach to analyze the uncertainties. (2) Supplementary Fig. 8 might give readers a wrong impression on the performance of abandonment maps of Crawford, et al. ¹¹. However, Supplementary Fig.14 only showcases two samples which are not representative to reflect the accuracy of their abandonment maps. More importantly, it is beyond our scope to assess the quality of their maps. At last, comparing our abandoned cropland map and Crawford, et al. ¹¹'s abandoned cropland map is not straightforward due to mismatch in spatial resolution.

What about the 1-year recultivation threshold, can this increase the risk of single-year pixel misclassifications affecting conclusions and contributing to the seen differences?

Response

We think it is less likely to cause a single-year pixel misclassification error. ESA CCI maps have

applied a three-year consistency rule, such that a detected land use change needs to be consistent for three consecutive years before appearing in the data (as you mentioned in the previous round of comments). This operation eliminates single-year pixel misclassifications. In other words, if a recultivation occurs, it will appear as cropland for at least three years.

We added a brief clarification of this issue in the Method section. Please see Line 503-505.

SI lines 390-400. I totally agree with this. It is mostly about having an open discussion that takes us closer to our common goal of understanding abandonment, recultivation, and land use change. And these previous studies are important.

Response

Thank you!

SI lines 400-405. I don't think that Supplementary Fig15 completely supports the conclusion you make here. A map of abandoned cropland extent in hectare (Supplementary Fig15a) is compared to locational proportions of abandoned land recultivated by 2017 (Supplementary Fig15b). Shouldn't hectares be compared with hectares and proportional recultivation with proportional recultivation? Of course, whilst clearly referring to the methodological differences in evaluated period, resolution etc. as already done.

Response

We agree with your comment. We used the location proportions of abandoned cropland recultivated by 2017 in Supplementary Fig. 15b. This was because the source data was obtained from Fig. S4 of Crawford, et al. ¹¹, where only recultivation proportions are available.

We deleted this part of the discussion to avoid unfair comparison.

Comment #22

Supplementary Text 8. The mapped historical recultivation seems like a key input to the approach that is used to quantify recultivation suitability. Based on the highlighted importance of methodological choices on quantified recultivation, I was hoping that you could reflect on how the use of a different method/dataset might in turn affect your results. Would it be likely to have a major impact, or can it be expected that the main conclusions would not change?

Response

Thank you for your suggestion. Our original model used 2.8 Mha (about 300,000 pixels) of historically recultivated abandoned cropland as input data to train a MaxEnt model and map abandoned cropland that is suitable for recultivation (Supplementary Fig. 5).

We designed a sensitivity analysis to reflect how the input historical recultivation map would in turn, affect the modeled suitability of recultivation. We randomly selected 95%, 90%, 10% of

historically recultivated abandoned cropland as input training data and analyzed how the resulting extent of abandoned cropland that is suitable for recultivation would be affected.

Figure R4. Relationship between the percentage of selected input data (historically recultivated abandoned cropland) and the resulting extent of abandoned cropland suitable for recultivation.

Our analysis indicates that as long as there is a sufficient number of input data (historical recultivated abandoned cropland pixels), the resulting extent of abandoned cropland suitable for recultivation is relatively stable (Supplementary Fig. 15).

Please also see the same figure at **Supplementary Text 9**.

Comment #23

Potential minor typos:

Line 342. “expercience”.

SI line 91. “when model when modelling[...]”.

Response

Thank you for pointing out these typos, we corrected all of them.

RESPONSE TO REVIEWER #2

Comment

I find that the authors adequately addressed all of my concerns. The added clarity around methods and figures leaves me with no remaining questions. Thank you for taking all of my comments into account, and especially for adjusting the conclusions regarding food provisioning.

Response

Thank you again for your encouraging words and insightful comments, which helped a lot in improving our manuscript.

RESPONSE TO REVIEWER #3

Comment

Thanks to the authors for their efforts. I am satisfied with the revision and recommend that it be published in its current form.

Response

Thank you again for your encouraging words and insightful comments, which helped a lot in improving our manuscript.

References

- 1 Xu, S. *et al.* Delayed use of bioenergy crops might threaten climate and food security. *Nature* **609**, 299-306 (2022).
- 2 Rosenzweig, C. *et al.* Assessing agricultural risks of climate change in the 21st century in a global gridded crop model intercomparison. *Proceedings of the National Academy of Sciences of the United States of America* **111**, 3268-3273, doi:10.1073/pnas.1222463110 (2014).
- 3 Liu, X. *et al.* Comparison of country-level cropland areas between ESA-CCI land cover maps and FAOSTAT data. *International Journal of Remote Sensing* **39**, 6631-6645, doi:10.1080/01431161.2018.1465613 (2018).
- 4 Lu, M. *et al.* A comparative analysis of five global cropland datasets in China. *Science China Earth Sciences* **59**, 2307-2317, doi:10.1007/s11430-016-5327-3 (2016).
- 5 Friedl, M. A. *et al.* MODIS Collection 5 global land cover: Algorithm refinements and characterization of new datasets. *Remote sensing of Environment* **114**, 168-182 (2010).
- 6 Hong, C. *et al.* The role of harmonized Landsat Sentinel-2 (HLS) products to reveal multiple trajectories and determinants of cropland abandonment in subtropical mountainous areas. *J Environ Manage* **336**, 117621, doi:10.1016/j.jenvman.2023.117621 (2023).
- 7 Yin, H. *et al.* Monitoring cropland abandonment with Landsat time series. *Remote Sensing of Environment* **246**, doi:10.1016/j.rse.2020.111873 (2020).
- 8 Biswas, W. K., Barton, L. & Carter, D. Global warming potential of wheat production in Western Australia: a life cycle assessment. *Water and Environment Journal* **22**, 206-216, doi:10.1111/j.1747-6593.2008.00127.x (2008).
- 9 Fischer, G. *et al.* Global Agro-ecological Zones (GAEZ v4)-Model Documentation. (2021).
- 10 Hempel, S., Frieler, K., Warszawski, L., Schewe, J. & Piontek, F. A trend-preserving bias correction – the ISI-MIP approach. *Earth System Dynamics* **4**, 219-236, doi:10.5194/esd-4-219-2013 (2013).
- 11 Crawford, C. L., Yin, H., Radeloff, V. C. & Wilcove, D. S. Rural land abandonment is too ephemeral to provide major benefits for biodiversity and climate. *Science Advances* **8**, eabm8999, doi:doi:10.1126/sciadv.abm8999 (2022).

Reviewers' Comments:

Reviewer #1:

Remarks to the Author:

All my previous comments have been convincingly addressed. The study has reached publication fitness. I congratulate the authors with their impressive revision efforts.

Final minor comments:

A couple of paragraphs are very long. For example, lines 414-459. I think it could be beneficial for readers if it was split into two or more shorter paragraphs.

Lines 55-58. The case made that previous assessments have been primarily single-purpose should be better backed up by references.

Jan Sandstad Næss

Response to Reviewer #1

Comment

All my previous comments have been convincingly addressed. The study has reached publication fitness. I congratulate the authors with their impressive revision efforts.

Thank you.

Response

We thank you for your insightful comments for all three rounds of review. We have well addressed your final comments as follows.

Final minor comments:

A couple of paragraphs are very long. For example, lines 414-459. I think it could be beneficial for readers if it was split into two or more shorter paragraphs.

Response

We have modified long paragraphs by splitting them into shorter ones.

Lines 55-58. The case made that previous assessments have been primarily single-purpose should be better backed up by references.

Response

We have added supporting references.